# LARGE LANGUAGE MODELS ARE ACTIVE CRITICS IN NLG EVALUATION

## ABSTRACT

The conventional paradigm of using large language models (LLMs) for evaluating natural language generation (NLG) systems typically relies on two key inputs: (1) a clear definition of the NLG task to be evaluated and (2) a list of pre-defined evaluation criteria. This process treats LLMs as "passive critics," strictly following human-defined criteria for evaluation. However, as new NLG tasks emerge, the criteria for assessing text quality can vary greatly. Consequently, these rigid evaluation methods struggle to adapt to diverse NLG tasks without extensive prompt engineering customized for each specific task. To address this limitation, we introduce **ACTIVE-CRITIC**, a novel LLM-based NLG evaluation protocol that enables LLMs to function as "active critics." Specifically, our protocol comprises two key stages. In the first stage, the LLM is instructed to infer the target NLG task and establish relevant evaluation criteria from the data. Building on this self-inferred information, the second stage dynamically optimizes the prompt to guide the LLM toward more human-aligned scoring decisions, while also generating detailed explanations to justify its evaluations. Experiments across four NLG evaluation tasks show that our approach achieves stronger alignment with human judgments than state-of-the-art evaluation methods. Our comprehensive analysis further highlights the effectiveness and explainability of ACTIVE-CRITIC with only a small amount of labeled data. We will share our code and data on GitHub.

## 1 INTRODUCTION

Recent advances in language technologies have catalyzed the development of natural language generation (NLG) systems, benefiting a variety of downstream applications such as text summarization (Fabbri et al., 2021), dialogue generation (Mehri & Eskenazi, 2020), and storytelling (Guan et al., 2021). However, despite the rapid growth of NLG systems, reliable techniques for automatic NLG evaluation still lay far behind, primarily due to the inherent challenges posed by the open-ended nature of NLG and the diverse demands of different stakeholders. This gap, in return, undermines the dependability of machine-based generators in real-world deployment.

Traditional NLG evaluation methods typically center on a specific criterion and require human-written references for comparison (Li et al., 2024). Commonly considered criteria include similarity-based reference alignment, text fluency, human likeness, and information adequacy. Some NLG tasks, such as text summarization, incorporate additional criteria like coherence and consistency (Lin & Chen, 2023). Notably, the measurement of alignment between machine-generated candidates and human-written references has been widely designed into various metrics, ranging from n-gram matching like BLEU (Papineni et al., 2002) and ROUGE (Lin, 2004) to embedding-based matching such as BERTScore (Zhang et al., 2019) and BARTScore (Yuan et al., 2021). To save the efforts of picking multiple appropriate single-aspect evaluators for all criteria regarding a specific NLG task, recent advent of large language models (LLMs) has introduced a new evaluation paradigm that is capable of scoring machine-generated candidates across multiple criteria simultaneously for diverse NLG tasks, either by fine-tuning the LLMs (Zhong et al., 2022; Jiang et al., 2023; Xu et al., 2023; Ke et al., 2023) or by prompting the model for assessment (Chiang & Lee, 2023; Gong & Mao, 2023; Lin et al., 2023). To save the high cost of human annotation on references and avoid potential biases caused by limited references, some latest works further emphasize designing reference-free evaluation methods (Fu et al., 2024; Liu et al., 2023a; Li et al., 2023; Jia et al., 2023).

Despite remarkable contributions made by prior work, existing NLG evaluation approaches commonly require two key inputs: (1) a clear *definition of the target NLG task* to be evaluated, and (2) a set of *predefined evaluation criteria* (e.g., coherence, relevance) to guide the machine evaluator's judgments. Shaped by the developers' understanding and expectations of the evaluation process, these machine evaluators often function as **"passive critics"**, constrained by the fixed criteria established by the developers. This limitation may hinder the machine from discovering valuable insights within the data that could enhance its assessment capabilities. Such constraints become even more pronounced when new NLG tasks are introduced, as the criteria for evaluating text quality can vary widely. As a result, these rigid evaluation methods struggle to adapt to diverse NLG tasks without extensive prompt engineering tailored to each specific task.

In this study, we concentrate on the potential of LLMs to evolve into **"active critics"**, motivated by the impressive performance of recent LLMs in in-context few-shot learning via prompting. We introduce ACTIVE-CRITIC—a novel LLM-based NLG evaluation protocol that assesses NLG systems based solely on the model's active engagement with data (i.e., machine-generated responses and their human-annotated scores). By automatically inferring what and how to evaluate from a few labeled data, this protocol offers flexibility in assessing diverse NLG tasks, particularly when adapting to new ones. Furthermore, it shows promise in capturing the diverse requirements of different stakeholders through their annotated data, facilitating more personalized evaluations.

To build ACTIVE-CRITIC, we develop a multi-stage evaluation pipeline that directs an LLM to (1) infer the target NLG task and identify relevant evaluation criteria from the data, and (2) optimize prompts to make its scoring better align with human judgments. To enhance the trustworthiness of ACTIVE-CRITIC, we also prompt the model to generate explanations alongside its scoring. Experiments on four NLG tasks using an open- and a closed-source LLM demonstrate that the ACTIVE-CRITIC outperforms various state-of-the-art baselines. Additionally, we show that ACTIVE-CRITIC is helpful in identifying more fine-grained evaluation aspects compared to developer-defined criteria, and its scoring explanations at both the aspect and overall levels can provide valuable insights. In summary, our contributions include:

**A novel NLG evaluation protocol**: ACTIVE-CRITIC functions as an active critic by automatically inferring the target task, establishing necessary evaluation criteria, refining its scoring, and generating fine-grained explanations to support its decisions.

**Extensive experiments across multiple NLG tasks**: We conduct extensive experiments to demonstrate the effectiveness and trustworthiness of ACTIVE-CRITIC in diverse NLG tasks.

**Comprehensive analysis and findings:** We show that combining fine-grained, criteria-specific scoring with explanation generation encourages the LLM to engage more deeply with the test cases, resulting in improved overall quality assessments.

## 2  RELATED WORK

**NLG Evaluation.** The current landscape of NLG evaluation methods can be viewed from two key perspectives. First, in terms of methodological design, these methods can be categorized into three groups, early human-centric evaluation (Mellish & Dale, 1998), followed by untrained machine evaluation (Papineni et al., 2002; Lin, 2004; Lavie & Denkowski, 2009), and more recently, machine-learned evaluation (Sennrich et al., 2015; Zhang et al., 2019; Yuan et al., 2021; Kim et al., 2023). Second, with respect to functionality, existing studies largely concentrate on single-criteria metric design, targeting either general NLG tasks like reference alignment (Liu et al., 2023b) or a specific NLG task like coherence for text summarization (Wang et al., 2023b). To enhance evaluation efficiency, some latest works have advocated for unified evaluation frameworks built on top of a pre-trained language model, aiming to transcend task-specific boundaries and assess multiple criteria simultaneously (Chiang & Lee, 2023; Liu et al., 2024; Gong & Mao, 2023). Since our work falls into this group, we discuss the details of these works below.

Overall, the study of unified evaluation frameworks encompasses two major strands. The first strand emphasizes the generality of evaluation methods, specifically targeting the estimation of instance quality scores based on defined tasks and criteria, w/wo human-written references (Xiao et al., 2023; Gao et al., 2024). Within this strand, researchers concentrate on two major strategies: (1) developing criteria-centered prompts that guide LLMs for multi-faceted, train-free evaluations (Fu et al., 2024;

Liu et al., 2023a; Lin & Chen, 2023), and (2) curating a large-scale multi-scenario benchmark to fine-tune an LLM as a generalized evaluator (Zhong et al., 2022; Li et al., 2023; Wang et al., 2023a; Ke et al., 2023). Comparatively, the first strategy offers a cost-effective approach to evaluation, while the second enhances scoring consistency and reproducibility. Moving beyond generality, the second strand of research focuses on evaluation interpretability, particularly in analyzing prediction errors by prompts (Xu et al., 2023; Jiang et al., 2023). In this study, we address both aspects of evaluation by introducing a new paradigm that guides the model to self-infer the target task and relevant evaluation criteria, ultimately making a final decision with free-text explanations drawn from several rated data instances, enabling the machine to be more active and flexible in evaluation.

**Dynamic Prompt Optimization.** Dynamic prompt optimization focuses on iteratively adjusting prompts to improve the performance of static LLMs on specific tasks. Existing methods can be categorized by inference depth into two groups: single-layer and multi-layer prompt optimization. Single-layer methods, such as APE (Zhou et al., 2023), APO (Pryzant et al., 2023), OPRO (Yang et al., 2023), and IPC (Levi et al., 2024), focus on optimizing prompts within a single stage, which limits their adaptability to complex tasks. In contrast, multi-layer methods like DSPy (Khattab et al., 2023) and MIPRO (Opsahl-Ong et al., 2024) optimize prompts across multiple stages, enabling more comprehensive reasoning but relying on scalar-based comparisons between data points, which fall short for tasks requiring correlations across data vectors. Our approach introduces a correlation-based comparison instead of a scalar-based comparison optimizing multi-stage NLG evaluation tasks. Specifically, we redesigned DSPy's prompt optimization algorithm using a two-stage process of Task Inference and Self-Optimizing Scoring, incorporating correlation-based validation metrics.

## 3 PRELIMINARY

**Problem Definition.** We target an LLM-based reference-free NLG evaluation task where ground-true human-written references are not provided. We are given a training dataset of $N$ examples $\mathcal{D}_{\text{train}} = \{(x_i, y_i, r_i)\}_{i=1}^N$, where $x_i$ is the $i$-th input text from the original NLG task, $y_i$ is the $i$-th output text of an NLG system, and $r_i \in \mathbb{R}$ is a numerical score measuring the output quality by humans. We denote the process of sampling a response from an LLM given a prompt as LLM([Prompt]) → [Response]. Our goal is to develop an explainable LLM critic that can automatically estimate a quality score $\hat{r}_i$[1] and a corresponding text explanation $e_i$ for an input-output pair $x_i, y_i$. That is, LLM$(x_i, y_i) \to \hat{r}_i, e_i$. We optimize the LLM critic such that their estimated scores strongly correlate with human judgments in $\mathcal{D}_{\text{train}}$, and evaluate the LLM critic on a hold-out test set $\mathcal{D}_{\text{test}}$.

**Prior Work.** Existing prompt-based methods using LLMs for evaluation often rely on manually designed rule-based criteria $c_{\text{rule}}$ (e.g., coverage, conciseness) tailored to a particular NLG task (e.g., text summarization), and incorporate these criteria into the prompt as LLM$(x_i, y_i, c_{\text{rule}}) \to \hat{r}_i$ without explanations. These methods face two major limitations. First, their rule-based criteria may not generalize well across different NLG evaluation tasks. Second, they require human effort to design criteria that align with the considerations human critics use for quality ratings. Consequently, verifying these rule-based criteria against the dataset $\mathcal{D}_{\text{train}}$ passively is often considered an afterthought.

## 4 ACTIVE-CRITIC

**Overview.** Our ACTIVE-CRITIC is a multi-stage evaluation framework designed for task-adaptive, explainable NLG evaluation. Unlike prior methods that use LLMs as a *passive critic*, our method optimizes the LLM as an *active critic* by searching for an optimal evaluation protocol $\Phi^*$ from the data itself that maximizes the correlations between the estimated ratings and human ratings in $\mathcal{D}_{\text{train}}$.

Specifically, we design two main stages: (1) *task inference* (§4.1) that obtains the text description $I$ of the NLG task and its key evaluation criteria; and (2) *self-optimizing scoring* (§4.2) that automatically derives the optimal few-shot examples $\mathcal{D}_{\text{demo}}$ from $\mathcal{D}_{\text{train}}$ for assessment. Figure 1 displays a workflow of these two stages, and Appendix A shows an example of the evaluation protocol $\Phi$ and its corresponding prompt $z$. Particularly, we implement the above optimization problem using

---

[1]The LLM critic outputs a text string of a numerical score, which can be further cast into the score $\hat{r}_i \in \mathbb{R}$.

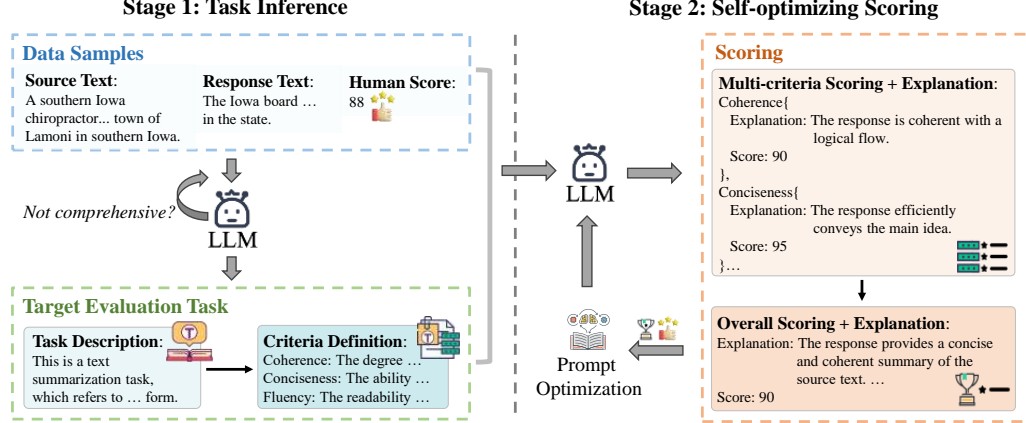

Figure 1: Overview of ACTIVE-CRITIC, including two stages: (1) task inference, where the LLM is instructed to derive the target NLG evaluation task description and relevant criteria from data samples, and (2) self-optimizing scoring, allowing the LLM to generate multi-criteria and overall quality scores along with accompanying explanations.

DSPy (Khattab et al., 2023), a declarative programming framework, as it offers convenient declarative modules to define each stage of our evaluation process.

## 4.1 TASK INFERENCE

The task inference stage, depicted on the left side of Figure 1, focuses on identifying two key components for NLG evaluation: (1) *task description* and (2) *criteria definition.* This stage aims to allow the LLM to autonomously analyze the input dataset, infer the characteristics of the NLG task, and establish relevant evaluation criteria without human intervention.

**Task Description.** This module focuses on prompting the LLM to formulate an accurate task description $T$ by reviewing examples in $\mathcal{D}_{\text{train}}$ and identifying key information that characterizes the target NLG task (e.g., summarization, storytelling) for evaluation. Considering that LLM's context length limit may not fit in all examples in $\mathcal{D}_{\text{train}}$, we split these examples into $N$ mini-batches, and generate one task description $T_n$ from each mini-batch $\mathcal{D}_{\text{train},n}$. That is, $\text{LLM}(f_t(\mathcal{D}_{\text{train},n})) \rightarrow T_n, \forall n \in [1, N]$, where $f_t$ is a prompt template shown in Table 9 in the appendix. The final task description $T$ is generated by the LLM through the ensemble of all task descriptions $\{T_n\}_{n=1}^N$ over all mini-batches.

**Criteria Definition.** After establishing the task description for each mini-batch, the LLM is instructed to define task-specific evaluation criteria for assessing the quality of machine-generated texts. Unlike traditional evaluation frameworks that rely on predefined criteria (e.g., coherence, fluency), we instruct the LLM to automatically identify the most relevant evaluation dimensions for the target NLG task. Similar to the task description, the final criteria set $C = \{c_1, c_2, ..., c_m\}^2$ is composed of all relevant dimensions inferred from all the reviewed mini-batches.

To enhance efficiency, we instruct the LLM to decide whether to stop early based on the comprehensiveness of the generated task description and criteria set after processing each mini-batch.

## 4.2 SELF-OPTIMIZING SCORING

Our second stage, as shown on the right side of Figure 1, focuses on automatically optimizing evaluation prompts to ensure the model delivers detailed, explainable scoring results that closely align with human judgments. Inspired by prior research that harnesses the potential of LLMs by breaking down complex tasks into simpler ones (Wei et al., 2022; Khot et al., 2023), we hypothesize that starting with fine-grained, criteria-specific scoring can help the model derive an accurate overall

---

[2]We instruct the LLM to output a criteria set in the JSON format, as shown in Table 10 in the appendix.

quality score. With this intuition in mind, we structure the scoring stage into two parts: (1) *Multi-criteria Scoring with Explanation (McS-E)*, followed by (2) *Overall Scoring with Explanation (OS-E)*. The prompts $z$ used in each component are treated as optimization parameters, and these prompts are template-based, generated from the evaluation protocol $\Phi$.

**Multi-criteria Scoring with Explanation (McS-E).**    In this module, the LLM assesses the model output $y_i$ based on the criteria set $C = \{c_1, c_2, \ldots, c_m\}$ obtained from the *task inference* stage (§4.1). Specifically, for each input-output pair $(x_i, y_i)$, the LLM is prompted to estimate a score $\hat{r}_{ij}$ and a corresponding explanation $e_{ij}$ according to each criterion $c_j \in C$:

$$\text{LLM}(x_i, y_i, f_{\text{McS-E}}(T, C, \mathcal{D}_{\text{demo}})) \to R_i = \{(\hat{r}_{ij}, e_{ij}), \forall c_j \in C\} \tag{1}$$

where the output uses a JSON format, indicating a set of score-explanation pairs $R_i$ for all criteria in $C$ and $\mathcal{D}_{\text{demo}}$ is a set of demonstration examples randomly selected from the training set $\mathcal{D}_{\text{train}}$. This mechanism ensures that the evaluation is both quantitative and interpretive, offering insights into the rationale behind each score. The prompt template $f_{\text{McS-E}}(T, C, \mathcal{D}_{\text{demo}})$ is designed to enable scoring across multiple criteria simultaneously, accounting for the interconnections between them. This design allows for a fine-grained evaluation, where each criterion is treated both individually and in connection with the others, providing detailed explanations that enhance the transparency and interpretability of the scoring process.

**Overall Scoring with Explanation (OS-E).**    After scoring the individual criteria, we use a prompt template $f_{\text{OS-E}}$ to instruct the LLM to synthesize these scores $\{\hat{r}_{i1}, \ldots \hat{r}_{im}\}$ into an overall quality score $\hat{r}_i$, and an explanation $e_i$ that provides a comprehensive justification for the final decision.

$$\text{LLM}(x_i, y_i, f_{\text{OS-E}}(T, R_i, \mathcal{D}_{\text{demo}})) \to \hat{r}_i, e_i \tag{2}$$

**Prompt Optimization.**    Given the sensitivity of LLM performance to the few-shot demonstration examples $\mathcal{D}_{\text{demo}}$ in the prompt, we further propose an automatic prompt optimization strategy built upon DSPy (Khattab et al., 2023) to iteratively select the optimal $\mathcal{D}_{\text{demo}}$ to refine the prompts. Specifically, given two lists of overall quality scores across all examples in $\mathcal{D}_{\text{train}}$—one predicted by the LLM ($\hat{R} = \{\hat{r}_i\}_{i=1}^N$ from Eq. 2) and the other annotated by humans ($R$)—we design an objective function to maximize the correlation between these two score lists. To mitigate potential biases caused by relying on a single correlation measurement, we calculate the sum of three widely-used correlation coefficients: Pearson ($\gamma$), Spearman ($\rho$), and Kendall ($\tau$) with equal weights:

$$\mathcal{Q}(\hat{R}, R) = \gamma(\hat{R}, R) + \rho(\hat{R}, R) + \tau(\hat{R}, R) \tag{3}$$

$$\Phi^* = \arg\max_{\Phi} Q(\hat{R}, R) \tag{4}$$

where $\Phi = (T, C, \mathcal{D}_{\text{demo}})$ is an evaluation protocol consisting of a text description of the NLG task $T$, evaluation criteria $C$, and a few demonstration examples $\mathcal{D}_{\text{demo}}$ randomly selected from $\mathcal{D}_{\text{train}}$. To approximately solve the above maximization problem, we repeat $K$ time for the evaluations of Eq. 2 using different randomly sampled $\mathcal{D}_{\text{demo}}$, and select the best $\mathcal{D}_{\text{demo}}$ that maximizes $Q(\hat{R}, R)$.

## 5 EXPERIMENT SETTINGS

**Benchmarks**    Following prior work (Zhong et al., 2022; Fu et al., 2024; Liu et al., 2023a), we employ four popularly-used benchmarks for meta-evaluation. These datasets cover diverse topics (e.g., politics, sports, restaurants, etc.) across four NLG tasks (i.e., summarization, dialogue generation, data-to-text generation, and storytelling), aiming to construct a robust testbed to access ACTIVE-CRITIC. The details of each benchmark are described below.

- **SummEval** (Fabbri et al., 2021) consists of 1,600 machine-generated summaries based on CN-N/DailyMail articles curated by (Hermann et al., 2015). Each summary is annotated by both expert and layman judges on four aspects: coherence, consistency, fluency, and relevance. The dataset provides both aspect-specific scores and an overall quality score.
- **Topical-Chat** (Mehri & Eskenazi, 2020) is a knowledge-grounded, open-domain conversation benchmark. It contains 60 conversations, each paired with 6 responses generated by different

systems (2 from humans and 4 from machines). Each dialogue response is evaluated with human scores on overall quality, considering five aspects: naturalness, coherence, engagingness, groundedness, and understandability.

- **SFRES** (Wen et al., 2015) is a data-to-text generation benchmark containing 1,181 instances. The task focuses on generating natural language utterances from structured restaurant information in San Francisco. The overall quality score is annotated by humans based on two aspects: informativeness and naturalness.

- **OpenMEVA (ROC)** (Guan et al., 2021) collects open-ended commonsense stories generated by various models trained upon ROCStories corpus. It includes 1,000 data instances in total. Each story is rated by humans on its overall quality regarding three aspects: fluency, creativity, and coherence.

We standardize all benchmarks into a uniform format that includes: (1) the machine-generated texts to be evaluated, (2) the source input used by generation systems to produce these texts, and (3) the human scores assessing the generated outputs' quality.

**Baselines and Metrics**    We compare ACTIVE-CRITIC with a variety of state-of-the-art publicly accessible NLG evaluation methods. The baselines are grouped into two categories: (1) fine-tuned models including Auto-J (Li et al., 2023) and UniEval (Zhong et al., 2022); and (2) prompting-based, train-free methods, including GPTScore (Fu et al., 2024), ExplainEval (Mahmoudi, 2023), and G-eval (Liu et al., 2023a). To ensure a fair comparison between the train-free baselines and ACTIVE-CRITIC, we use the same backbone LLM in comparisons. For GPTScore, we denote GPTScore-src to indicate the use of the src-hypo scoring type.

We focus on three correlation coefficients to assess the evaluation consistency between machine-based evaluators and humans: Pearson ($\gamma$) (Mukaka, 2012), Spearman ($\rho$) (Zar, 2005) and Kendall-Tau ($\tau$) (Kendall, 1938).

**Meta-evaluation**    We establish ACTIVE-CRITIC using two widely adopted backbone models: one open-source LLM—Orca2-13b, and the other close-source LLM—GPT-3.5 (gpt-3.5-turbo-1106). In our preliminary study, we also tested ACTIVE-CRITIC on LLaMA2-13B. Since Orca2-13B outperformed LLaMA2-13B in achieving higher alignment with human judgments, we selected Orca2-13B for further analysis. Given the long-term accessibility and lower costs of open-source LLMs compared to closed-source models, we focus on Orca2-13B-based ACTIVE-CRITIC across four NLG tasks. For comparison, we built the GPT-3.5-based ACTIVE-CRITIC with a specific focus on the two most commonly used NLG tasks (SummEval and TopicalChat) in the meta-evaluation.

We focus on three variants of ACTIVE-CRITIC (AC) in the meta-evaluation. First, **AC-Vanilla** is a protocol that directly prompts the base LLM to score the overall quality of the input test case using self-optimizing scoring. Without providing or guiding the LLM to generate task-specific information, this variant explores the base LLM's ability to make judgments based solely on its intrinsic understanding of few-shot graded exemplars. The second variant **AC-Coarse**, performs a coarse-grained, explainable evaluation by prompting the LLM to infer task-specific information (i.e., task definition and necessary criteria) and produce an overall score along with an explanation for each test case. This process considers all inferred criteria simultaneously during the self-optimizing scoring. Finally, **AC-Fine** provides a fine-grained, explainable evaluation. Similar to AC-Coarse, it begins with task inference, but during self-optimizing scoring, it assesses the input test case against each criterion individually, offering detailed explanations for each score. The overall quality score is then generated by combining the evaluations across all criteria. Details of the parameter settings and implementation are provided in Appendix B.

## 6    RESULTS AND ANALYSIS

### 6.1    OVERVIEW OF ACTIVE-CRITIC PERFORMANCE

**ACTIVE-CRITIC outperforms baseline evaluators across four distinct NLG tasks.**    Table 1 displays the correlation results between unified evaluators and human judgments on four distinct NLG tasks. Overall, we observe that the coarse and fine variants of ACTIVE-CRITIC consistently exhibit

| | SummEval | | | TopicalChat | | | SFRES | | | OpenMEVA (ROC) | | | Average |
|---|---|---|---|---|---|---|---|---|---|---|---|---|---|
| | $\gamma$ | $\rho$ | $\tau$ | $\gamma$ | $\rho$ | $\tau$ | $\gamma$ | $\rho$ | $\tau$ | $\gamma$ | $\rho$ | $\tau$ | |
| Auto-J | 0.1345 | 0.1457 | 0.1149 | 0.4681 | 0.459 | 0.3714 | 0.126 | 0.1022 | 0.0809 | 0.3896 | 0.3704 | 0.3065 | 0.2558 |
| UniEval | 0.5457 | 0.4914 | 0.3707 | 0.5133 | 0.5448 | 0.4134 | 0.2894 | 0.2499 | 0.1877 | 0.4501 | 0.4408 | 0.3119 | 0.4008 |
| GPTScore-src | 0.4043 | 0.3584 | 0.2696 | 0.2313 | 0.2437 | 0.1792 | 0.2062 | 0.121 | 0.1132 | 0.2283 | 0.2265 | 0.1534 | 0.2280 |
| ExplainEval | 0.5447 | 0.4916 | 0.3999 | 0.5542 | 0.5512 | 0.4476 | 0.1993 | 0.1181 | 0.0946 | 0.4809 | 0.4695 | 0.358 | 0.3924 |
| **Ours** | | | | | | | | | | | | | |
| AC-VANILLA | 0.5494 | 0.486 | 0.3964 | 0.4932 | 0.4927 | 0.4055 | 0.2322 | 0.1712 | 0.1328 | 0.4023 | 0.4245 | 0.3209 | 0.3756 |
| AC-COARSE | 0.5386 | 0.5227 | 0.4156 | **0.611** | 0.6173 | **0.4845** | **0.3094** | **0.2663** | **0.199** | 0.4908 | 0.4962 | 0.3622 | 0.4428 |
| AC-FINE | **0.6301** | **0.5486** | **0.4299** | 0.6023 | **0.6214** | 0.4713 | 0.2915 | 0.2501 | 0.1906 | **0.5259** | **0.5363** | **0.4109** | **0.4591** |

Table 1: Correlation between LLM-based unified evaluators and human judgments on overall quality per instance across four NLG tasks. All train-free evaluators are built upon Orca2-13B. We compare Pearson ($\gamma$), Spearman ($\rho$) and Kendall-Tau ($\tau$) correlation, respectively. The best performance per indicator is highlighted in bold, and the second-highest results are underlined. We implemented and tested all the methods with p-value $< 0.05$.

a higher correlation with human judgments than baselines across three correlation coefficients in four NLG tasks. The fine-level evaluation generally outperforms the coarse variant. Among three variants, the vanilla ACTIVE-CRITIC performs the worst, even with lower average correlation compared to UniEval and ExplainEval. Our observations suggest that while the backbone LLM's (i.e., Orca2-13B) intrinsic knowledge is limited for directly scoring data without task-specific inference, even with prompt optimization using a few examples, our proposed active-critic mechanism can effectively unlock the model's potential to deeply digging into the example data for evaluation. Moreover, this mechanism enhances the model's ability to assess data quality with greater alignment to human judgments. Notably, guiding the model to evaluate each inferred criterion individually leads to better final decisions than asking it to directly provide an overall score.

**ACTIVE-CRITIC can achieve further improvements with a stronger backbone LLM.** Table 2 displays the results of ACTIVE-CRITIC built on top of GPT-3.5, with an emphasis on the task of SummEval and TopicalChat.

Comparatively, ACTIVE-CRITIC built on GPT-3.5 shows significantly higher correlations with human judgments than the variant built on Orca-13B, indicating that a stronger backbone model enhances the effectiveness of our evaluation protocol.

| | SummEval | | | TopicalChat | | |
|---|---|---|---|---|---|---|
| | $\gamma$ | $\rho$ | $\tau$ | $\gamma$ | $\rho$ | $\tau$ |
| G-eval | 0.4687 | 0.4504 | 0.3745 | 0.5427 | 0.5597 | 0.4501 |
| **Ours** | | | | | | |
| AC-Coarse | **0.6569** | 0.5368 | 0.4178 | 0.6425 | 0.6171 | 0.4855 |
| AC-Fine | 0.653 | **0.6016** | **0.4745** | **0.6718** | **0.6703** | **0.5156** |

Table 2: Correlation between GPT-3.5-based evaluators and human judgments on instance-level overall quality for SummEval and TopicalChat. We compare Pearson ($\gamma$), Spearman ($\rho$) and Kendall-Tau ($\tau$) correlation, respectively. The best performance per indicator is highlighted in bold, and the second-highest results are underlined. We implemented and tested all the methods with p-value $< 0.05$.

Further comparison with the state-of-the-art GPT-3.5-based baseline reveals that ACTIVE-CRITIC consistently outperforms it. As with the Orca-13B-based ACTIVE-CRITIC (see Table 2), the fine-level variant surpasses the coarse one, further suggesting that multi-criteria scoring and explanations help the backbone LLM conduct a more in-depth analysis of instance quality and make better final decisions.

## 6.2 EXPLAINABILITY ANALYSIS

**Are the generated explanations helpful to ACTIVE-CRITIC's scoring?** To assess the impact of explanations generated by ACTIVE-CRITIC, we compared our protocol's performance with versus without explanations, at both coarse and fine levels of evaluations. Figure 2 shows the results based on the Kendall-Tau correlation. We also provide the results of Pearson and Spearman correlation in Appendix D.

As shown in Figures 2, ACTIVE-CRITIC with explanations consistently demonstrates a higher correlation with human judgments than the version without explanations. Notably, the difference in correlation is greater for the fine-level ACTIVE-CRITIC compared to the coarse-level variant. These findings suggest that generating explanations for scoring helps the base LLM engage more effec-

| Dimension | Clarity | Relevance | Score Consistency | Accuracy | Aspect-to-Overall Alignment | Differentiability | Overall |
|---|---|---|---|---|---|---|---|
| Rate→ | Yes (%) | Yes (%) | Yes (%) | Yes (%) | Yes (%) | Yes (%) | (1-5) |
| Coherence | 99.11 | 92 | 95.78 | 85.33 | | | |
| Conciseness | 98.67 | 91.78 | 96.89 | 88.89 | | | |
| Coverage | 98.82 | 91.33 | 97.56 | 96.89 | | | |
| Accuracy | 98.22 | 92.22 | 95.56 | 98 | | | |
| Fluency | 99.56 | 98.89 | 96 | 96.67 | 95.11 | 90 | 4.515 |
| Relevance | 98.89 | 99.11 | 98.44 | 95.56 | | | |
| Clarity | 98 | 94.22 | 93.56 | 95.78 | | | |
| Engagement | 99.33 | 94.67 | 93.33 | 91.11 | | | |
| Overall Quality | 98.44 | 98.44 | 97.33 | 98 | | | |
| Average | 98.78 | 94.74 | 96.05 | 94.03 | - | - | - |

Table 3: Human evaluation of explanations generated by ACTIVE-CRITIC on SummEval samples. It assesses (1) the quality of individual explanations, (2) the alignment of the overall explanation with criteria-specific explanations, (3) the distinguishability of the overall explanation across cases of varying quality, and (4) the overall usefulness of the generated explanations per testing case.

tively in the evaluation process, resulting in stronger alignment with human judgments. In particular, fine-level explanations for each model-inferred criterion are especially effective in boosting the model's engagement and improving evaluation accuracy.

**Human Evaluation.** To conduct a deeper analysis of the explanations generated by ACTIVE-CRITIC, we employed three proficient English-speaking annotators to evaluate the quality of the scoring explanations on a random sample of 150 test cases from SummEval. Our assessment consisted of four parts, details in Appendix F. First, for each individual explanation per case, each annotator rated the quality based on: (1) clarity of the statement, (2) relevance to the target criterion, (3) alignment with the corresponding score, and (4) accuracy within the context of the test case (e.g., correctness in matching the source

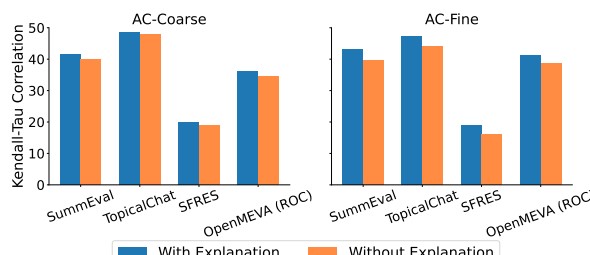

Figure 2: Comparison of the performance of ACTIVE-CRITIC w/ vs. w/o explanations across four NLG tasks. We measure ACTIVE-CRITIC's performance by Kendall-Tau correlation (%). Both coarse-level (AC-Coarse) and fine-level (AC-Fine) variants are investigated.

text). Further emphasizing the overall scoring explanation per case, we asked annotators to assess its alignment with the criteria-specific explanations, and its differentiability across cases of varying quality, respectively. Finally, we asked annotators to provide an overall rating on a scale of 1-5 based on the usefulness of all generated explanations per case. To validate the reliability of human annotations, following prior work (Fabbri et al., 2021), we calculated intercoder reliability by Krippendorff's alpha (Krippendorff, 2011). The 0.6534 Kappa coefficient indicates substantial agreement among annotators.

Table 3 shows the results. Overall, we observe a comparatively high quality of individual explanations over four considered dimensions, with 98.78% clarity, 94.74% relevance, 96.05% score consistency, and 94.03% information accuracy on average. Most testing cases have overall explanations that align with the criteria-specific ones (95.11%), and 90% of the overall explanations effectively differentiate case quality. With an average rating of ∼4.5 out of 5 on the generated explanations across sampled testing cases, the result shows that explanations generated by ACTIVE-CRITIC are of good quality and useful to explain the resulting scores.

## 6.3 ABLATION STUDY

**Impact of Optimization.** We assess the impact of optimization on ACTIVE-CRITIC by comparing its performance when dynamic prompt optimization for scoring is removed and, additionally, when mini-batch iterations are eliminated during task inference. Figure 3 displays the results. Across all four NLG tasks, we observe a consistent performance drop when prompt optimization for scoring is removed, with a further decline when only using a single mini-batch of labeled data for task inference. This suggests that both scoring prompt optimization and mini-batch iterations for task

| | SummEval | | | TopicalChat | | | SFRES | | | OpenMEVA (ROC) | | | Average |
|---|---|---|---|---|---|---|---|---|---|---|---|---|---|
| | $\gamma$ | $\rho$ | $\tau$ | $\gamma$ | $\rho$ | $\tau$ | $\gamma$ | $\rho$ | $\tau$ | $\gamma$ | $\rho$ | $\tau$ | |
| **Ours (AC-Fine)** | **0.6301** | **0.5486** | **0.4299** | 0.6023 | **0.6214** | 0.4713 | 0.2915 | 0.2501 | 0.1906 | **0.5259** | **0.5363** | **0.4109** | **0.4591** |
| w/o Task Description | 0.5825 | 0.4826 | 0.3552 | 0.4949 | 0.5057 | 0.4211 | 0.2011 | 0.1406 | 0.1129 | 0.3846 | 0.3802 | 0.2918 | 0.3628 |
| w/o Criteria Definition | 0.5726 | 0.522 | 0.4062 | 0.5533 | 0.5368 | 0.4451 | 0.2567 | 0.243 | 0.1571 | 0.4176 | 0.4237 | 0.326 | 0.405 |
| w/o McS-E | 0.5386 | 0.5227 | 0.4156 | **0.611** | 0.6173 | **0.4845** | **0.3094** | **0.2663** | **0.199** | 0.4908 | 0.4962 | 0.3622 | 0.4428 |
| w/o OS-E | 0.6106 | 0.5129 | 0.3908 | 0.5639 | 0.5615 | 0.4464 | 0.2844 | 0.2113 | 0.1602 | 0.509 | 0.4931 | 0.3632 | 0.4256 |

Table 4: Ablation study of key modules in ACTIVE-CRITIC.

inference are crucial for ACTIVE-CRITIC to achieve more human-aligned evaluations. Interestingly, the ACTIVE-CRITIC shows greater sensitivity to scoring optimization in the fine-level evaluation of SummEval and the coarse-level evaluation of SFRES, indicating that this component plays a more significant role in these specific evaluation scenarios. In contrast, the influence of mini-batch iterations for task inference is minimal in SummEval, suggesting that ACTIVE-CRITIC can effectively infer the target evaluation task in this setting with limited training data.

**Key Module Analysis.** We further analyze the individual contribution of each module in ACTIVE-CRITIC by comparing performance with and without that module. Table 4 shows the results across four NLG tasks. Noted that, the variant w/o criteria inference uses the original predefined criteria from each benchmark for further computation. In the variant w/o OS-E, we calculated the overall quality score per test case by averaging the multiple criteria-specific scores generated from McS-E.

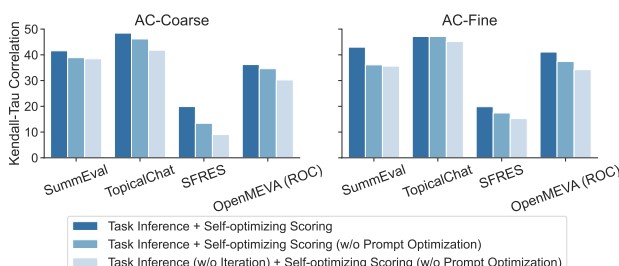

Figure 3: Impact of prompt optimization for scoring and mini-batch iterations for task inference. We compare ACTIVE-CRITIC's performance using Kendall-Tau(%). The Pearson and Spearman results are in Appendix E.

Overall, we find that removing task inference modules leads to a more substantial performance drop in our protocol compared to scoring modules, especially when the LLM is not asked to infer the task description (resulting in a ∼10% decrease on average). This suggests that our approach of guiding LLM in determining what to evaluate is more important to ACTIVE-CRITIC's effectiveness than other modules. The greater performance drop for the variant w/o OS-E, compared to the one w/o McS-E, indicates that the LLM-generated overall quality score contributes more meaningfully than simply averaging the generated criteria-specific scores.

# 7 CONCLUSION AND DISCUSSION

We proposed ACTIVE-CRITIC, a novel LLM-based NLG evaluation protocol that relies solely on lightweight human-scored data. Unlike existing machine-based evaluators that depend on human-predefined task descriptions and evaluation criteria, ACTIVE-CRITIC actively infers the necessary details about the target evaluation task and provides explainable assessments by drawing insights directly from the data. This paradigm shift will open the door to endow ACTIVE-CRITIC with the adaptability to evaluate systems on new NLG tasks without extensive task-specific prompt engineering. Experiments across four distinct NLG tasks demonstrate LLMs' potential as active critics, achieving a higher correlation with human judgments compared to state-of-the-art baseline evaluators. Fine-level criteria-specific scoring, paired with the explanation generation setting, prompts the LLM to engage more deeply with the test cases, leading to improved overall quality scoring.

Our work has several limitations. First, due to resource constraints, we primarily focused on four existing NLG tasks and benchmarks for meta-evaluation in our experiments. It would be valuable to deploy our protocol in a broader testing environment to assess its performance in more diverse settings. Additionally, building ACTIVE-CRITIC on a wider range of backbone LLMs could provide deeper insights. Overall, we hope this study will contribute to advancing generic NLG evaluation research and promote system development across diverse NLG scenarios.

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

## A    AN EXAMPLE OF EVALUATION PROTOCOL AND PROMPT ON SUMMEVAL

### A.1    AN EXAMPLE OF INPUT DATA

This section shows an example of data $(x_i, y_i, r_i)$ from SummEval in Table 5.

| **Source** $(x_i)$ |
|---|
| A southern Iowa chiropractor accused of accepting sex as payment for his services and performing exorcisms on patients has surrendered his state license. The Iowa Board of Chiropractic released a report Wednesday detailing charges against Charles Manuel, of Lamoni. Manuel signed an agreement last month admitting his misdeeds and pledging not to apply for reinstatement for at least 10 years. Patient satisfaction: A chiropractor in Iowa has surrendered his license to practice and admitted to swapping services for sex and performing exorcisms on some patients. Nonetheless, he's received outstanding evaluations for patient satisfaction on HealthGrades.com The agreement requires that Manuel prove any circumstances surrounding the revocation of his license no longer exist before he can resume practicing chiropractic in the state. Those circumstances included bartering sex for services with some patients. Manuel also recommended that patients stop taking medication he prescribed to them. A woman who answered a call to Manuel's home from the Des Moines Register declined to comment on the case. A woman at his former practice said he hadn't worked there for some time. A Lamoni address listed on the Yelp page for Manuel's practice appears to be a home on a residential street. While Maneul has received just three patient survey responses on HealthGrades.com, those responses were quite positive. The disgraced chiropractor received a perfect five out of five stars in patient satisfaction. Strange Practice: Charles Manuel, who admitted wrongdoing to the Iowa Board of Chiropractic, listed his practice's addresses on this residential street in the small agricultural town of Lamoni in southern Iowa. |
| **Response** $(y_i)$ |
| The iowa board of chiropractic released a report wednesday detailing charges against charles manual, of lamoni. Manuel signed an agreement last month admitting his misdeeds and pledging not to apply for reinstatement for at least 10 years . The agreement requires that manuel prove any circumstances surrounding the revocation of his license no longer exist before he can resume practicing chiropractic in the state. |
| **Human Judgment Score** $(r_i)$ |
| 88.33333333333332 |

Table 5: An example of data $(x_i, y_i, r_i)$ from SummEval

## A.2 AN EXAMPLE OF OUTPUT

This section shows an example of output with our AC-Fine method in table 6.

---

**Multi-Criteria Scoring with Explanation**

Coherence: {
  explanation: The response is coherent and logically flows from one point to the next, with clear connections between the main points.
  score: 90
}
Conciseness: {
  explanation: The response is concise and efficiently conveys the main events and details from the source text.
  score: 90
}
Coverage: {
  explanation: The response covers the main events and details from the source text without omitting crucial information.
  score: 85
}
Accuracy: {
  explanation: The response accurately reflects the main points and details of the source text.
  score: 90
}
Fluency:{
  explanation: The response is fluent, with smooth transitions between ideas and paragraphs, and natural language used throughout.
  score: 95
}
Relevance: {
  explanation: The response is highly relevant to the main topic and includes pertinent information from the source text.
  score: 95
}
Clarity: {
  explanation: The response is clear and comprehensible, with precise language used to convey the main points effectively.
  score: 90
}
Engagement: {
  explanation: The response effectively draws the reader into the main events and details, making it highly engaging.
  score: 95
}

---

**Overall Scoring with Explanation**

**Explanation:** The response provides a concise and coherent summary of the source text, capturing the main events and details without omitting crucial information. The summary is accurate, relevant, and clear, with smooth transitions between ideas and paragraphs, making it highly engaging and readable. The coverage of the main events and details is comprehensive, and the language used is clear and precise. Overall, the response effectively conveys the main points of the source text in a compelling and engaging manner.

**Overall Score:** 90.0

---

Table 6: An example of output with our AC-Fine method

## A.3 INSTRUCTION COMPONENT OF THE EVALUATION PROTOCOL

This section presents the instruction $I$ in evaluation protocol $\Phi$, which is also the output of the task inference module, in table 7

---

**Task Description $T$:**

The evaluation task is text summarization, which refers to evaluating abstractive summaries generated from source texts. The relationship between 'Source' and 'Response' is that the responses consistently provide a summary of the main events or details described in the source text and accurately reflect the main points of the source text in a summarized form.

**Critiria $C$:**

"Coherence": "The degree to which the summary flows logically and cohesively, with clear connections between the main points."
"Conciseness": "The ability of the summary to convey all necessary information in a succinct and efficient manner."
"Coverage": "The extent to which the summary captures the main events and details from the source text without omitting crucial information."
"Accuracy": "The faithfulness of the summary to accurately reflect the main points and details of the source text."
"Fluency": "The readability and naturalness of the language used in the summary, with smooth transitions between ideas and paragraphs."
"Relevance": "The relevance of the summary to the main topic and the inclusion of only pertinent information from the source text."
"Clarity": "The clarity and comprehensibility of the summary, with clear and precise language used to convey the main points."
"Engagement": "The ability of the summary to captivate and engage the reader, drawing them into the main events and details effectively."

---

Table 7: An example of instruction $I$ in evaluation protocol $\Phi$

## A.4 IN-CONTEXT EXEMPLAR OF THE EVALUATION PROTOCOL

This section presents the in-context exemplar $D_{demo}$ in evaluation protocol $\Phi$ in table 8

---

**AC-Fine Output Example:**

**Source:** "Paul Merson has restarted his row with Andros Townsend... Any bad feeling between the pair seemed to have passed but Merson was unable to resist having another dig at Townsend after Tottenham drew at Turf Moor."

**Response:** "Paul merson has restarted his row with andros townsend .. in the 83rd minutefor tottenham as they drew 0-0 against burnley."

**Multiple Evaluation Criteria:**

Coherence: The degree to which the summaryflows logically and cohesively, with clearconnections between the main point.

Conciseness: The ability of the summary to convey all necessaryinformation in a succinctand efficient manner.

...

**Score Of Each Criterion In JSON:**

Coherence: {

      Explanation: The response is somewhat coherent, but it jumps between different events and

      details without clear connections between them.

      Score: 60

      }

...

**Explanation:** The response provides a concise summary ... to provide a more compelling and logically flowing summary.

**Score of overall:** 75

---

$\mathcal{D}_{train}$ **Example:**

**Source:** Chelsea have made an offer for FC... The initial five-year deal is the biggest in the club 's history , with the Blues now considering a two-week pre-season tour of Japan this summer.

**Response:** Chelsea have made an offer for fc ... in muto is not connected to the 200million sponsorship deal they signed with japanese company yokohama rubber in February.

**"Score of Overall":** 91.66666666666666

---

Table 8: An example of in-context exemplar $D_{demo}$

## A.5 PROMPT TEMPLATE

This section presents prompt templates in multiple stages: (1) Task Description (Table 9), (2) Criteria Definition (Table 10), (3) Multi-Criteria Scoring with Explanation (Table 11), and (4) Overall Scoring with Explanation (Table 12).

---

Given several examples from an NLG evaluation dataset where each entry consists of a 'Source' text and its corresponding 'Response', along with a score that evaluates the response quality.
Please write observations about trends that hold for most or all of the samples.
I will also provide you with some previous observations I have already made. Please add your observations or if you feel the observations are comprehensive say 'COMPLETE'.
Some areas you may consider in your observations: content and structure, scenario, task, evaluation objective, evaluation criteria, etc.
It will be useful to make an educated guess as to the nature of the task this dataset will enable. Don't be afraid to be creative.
${*examples*}
${*prior observations*}

---

Given a series of observations I have made and some description about this NLG evaluation dataset.
    1. Identify the type of evaluation task. Possible tasks include: machine translation, text summarization, data-to-text generation, dialogue generation, image description, text simplification, story generation, paraphrase generation, textual entailment, reasoning, etc.
    2. What this evaluation task refers to evaluating.
    3. Output the relationship between 'Source' and 'Response' in this task in 1-3 sentences.
    4. Given a summary in fill [ ]: The evaluation task is [ ], which refers to evaluating [ ] generated from [ ]. The relationship between 'Source' and 'Response' is [ ].
${*observations*}
${*prior task description*}

---

Table 9: Prompt template on Task Description

---

Given a task description about this NLG evaluation dataset and a series of observations I have made.
Your task is to list ten aspects that can be considered when measuring the overall quality of ${*task type*}.
${*task description*}
${*observations*}
Output in JSON format: aspect as key, description as value.

---

From the provided sets of criteria for evaluating ${*task type*}, identify the key aspects that are essential for this task. Select between 4 to 10 criteria that best align with the goals of your evaluation task and prioritize them based on their importance to the overall quality of the ${*task type*}.
${*sets of criteria*}
Output in JSON format: aspect as key, description as value.

---

Table 10: Prompt template on Criteria Definition

${*Task Description*}
Your task is to evaluate the response on multiple evaluation criteria with respect to the source on a continuous scale from 0 to 100, and explain your process for scoring each criterion. Rate the response on multiple evaluation criteria and give a brief explanation in a JSON format by filling in the placeholders in [ ].

${*In-context exemplar*}

${*Source*}
${*Response*}
${*Multiple Evaluation Criteria*}

Output format:
Score Of Each Criterion In JSON:

{
Coherence: {
        Explanation: "[your explanation]",
        Score: "[score from 0 to 100: 0 - No logic, 100 - Perfectly coherent]" },
Conciseness: {
        Explanation: "[your explanation]",
        Score: "[score from 0 to 100: 0- Overly verbose, 100- Highly efficient]" },
...
}

Table 11: Prompt template on Multi-Criteria Scoring with Explanation

${*Task Description*}
Your task is to rate the overall quality of the response, based on the source and the scores for different criteria of the response on a continuous scale from 0 to 100, where 0 means 'completely irrelevant and unclear' and 100 means 'perfectly relevant, clear, and engaging.' IMPORTANT!! Only output the score as an 'int' and nothing else.
"Also explain your process to get this score to response. Also please perform error Analysis of given response. What should we change to have a better result?"

${*In-context exemplar*}

${*Source*}
${*Response*}
${*Score Of Different Criteria*}

Output format:

Explanation:
Score Of Overall:

Table 12: Prompt template on Overall Scoring with Explanation

## B  Details of Parameter Setting and Implementation

We randomly sample 25% of the data for ACTIVE-CRITIC tuning and use the remaining 75% for meta-evaluation across each NLG task. During task inference, we set the number of mini-batches to 25, with a batch size of 5. The LLM is instructed to generate one task description and a set of evaluation criteria per mini-batch. To enhance tuning efficiency, we allow the LLM to decide when to stop early, capping the number of task descriptions and criteria sets at 5. For the scoring stage, we run 11 epochs of prompt optimization. The number of in-context exemplars used per epoch is 3 for SummEval and TopicalChat, and 8 for SFRES and OpenMeVA (ROC), with the difference due to varying input text lengths across tasks. All parameter settings are based on empirical testing of sequential values to determine optimal configurations.

Our experiments were carried out using two NVIDIA V100 GPU cards. For prompt optimization in the scoring stage, we utilized the "BootstrapFewShotWithRandomSearch" method in DSPy (Khattab et al., 2023) as the optimizer, which leverages random search to generate examples.

## C  Qualitative Analysis of LLM-inferred Criteria.

To analyze the process of automatic evaluation criteria inference by ACTIVE-CRITIC in depth, we conducted a qualitative analysis by reading through three types of criteria per task: (1) criteria generated by AC-Fine, (2) criteria generated by AC-Fine w/o task description, and (3) criteria predefined by humans in each original benchmark. Table 13 shows an illustrative example of SummEval. Moving beyond four primary aspects considered by humans in summarization, ACTIVE-CRITIC also incorporates a range of nuanced criteria, such as clarity, conciseness, coverage/completeness, and engagement. Additionally, each criterion is clearly defined to specify its distinct characteristics.

| (a) AC-Fine | (b) AC-Fine w/o Task Description | (c) Human |
|---|---|---|
| **Coherence**: The degree to which the summary flows logically and cohesively, with clear connections between the main points. **Conciseness**: The ability of the summary to convey all necessary information in a succinct and efficient manner. **Coverage**: The extent to which the summary captures the main events and details from the source text without omitting crucial information. **Accuracy**: The faithfulness of the summary to accurately reflect the main points and details of the source text. **Fluency**: The readability and naturalness of the language used in the summary, with smooth transitions between ideas and paragraphs. **Relevance**: The relevance of the summary to the main topic and the inclusion of only pertinent information from the source text. **Clarity**: The clarity and comprehensibility of the summary, with clear and precise language used to convey the main points. **Engagement**: The ability of the summary to captivate and engage the reader, drawing them into the main events and details effectively. | **Relevance**: How well the response captures the key points or main events from the source text. **Conciseness**: "The extent to which the response is brief and to the point, focusing on essential information. **Factual Accuracy**: Ensuring that the response maintains accuracy and does not introduce new or incorrect information. **Clarity**: The clarity of language used in the response, avoiding ambiguity or confusion. **Structure**: The organization and coherence of the response, including the presence of an introductory sentence and logical flow. **Completeness**: Whether the response covers all the main highlights from the source text without omitting crucial information. **Coherence**: The overall coherence and logical progression of ideas in the response. **Engagement**: The ability of the response to engage the reader and maintain interest. **Novelty**: The extent to which the response introduces new insights or perspectives beyond the source text. **Consistency**: Ensuring that the responses exhibit a consistent style, tone, and level of detail throughout the dataset. | **Coherence**: the summary should be well-structured and well-organized. The summary should not just be a heap of related information, but should build from sentence to sentence to a coherent body of information about a topic. **Consistency**: the factual alignment between the summary and the summarized source. A factually consistent summary contains only statements that are entailed by the source document. **Fluency**: the summary should have no formatting problems, capitalization errors or obviously ungrammatical sentences (e.g., fragments, missing components) that make the text difficult to read. **Relevance**: The summary should include only important information from the source document. |

Table 13: An illustrative example of the generated evaluation criteria on SummEval, either generated by an ACTIVE-CRITIC variant (a & b) or predefined by humans (c).

# D    IMPACT OF EXPLANATIONS BY PEARSON AND SPEARMAN CORRELATION

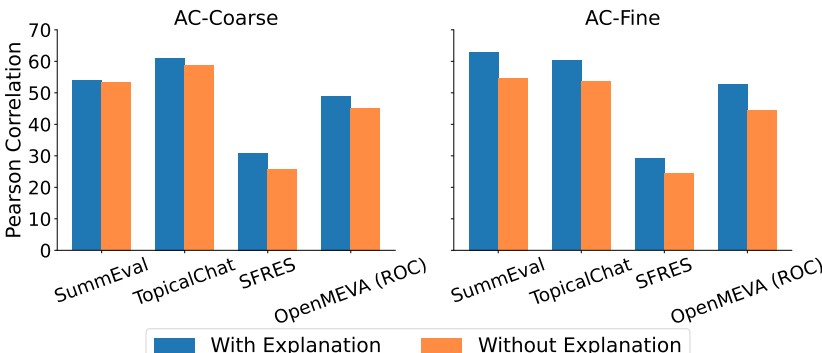

Figure 4: Effectiveness of Explanation in Pearson ($\gamma$).

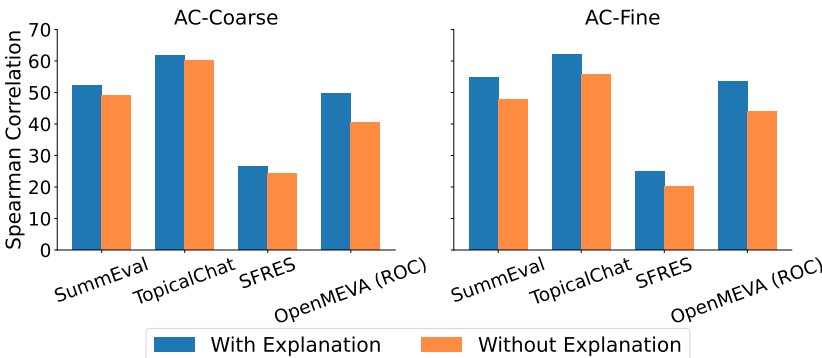

Figure 5: Effectiveness of Explanation in Spearman ($\rho$). We report the average on Spearman ($\rho$) correlation coefficients for our two variants: AC-Coarse, and AC-Fine, each presented with and without explanation.

# E    IMPACT OF OPTIMIZATION BY PEARSON AND SPEARMAN CORRELATION

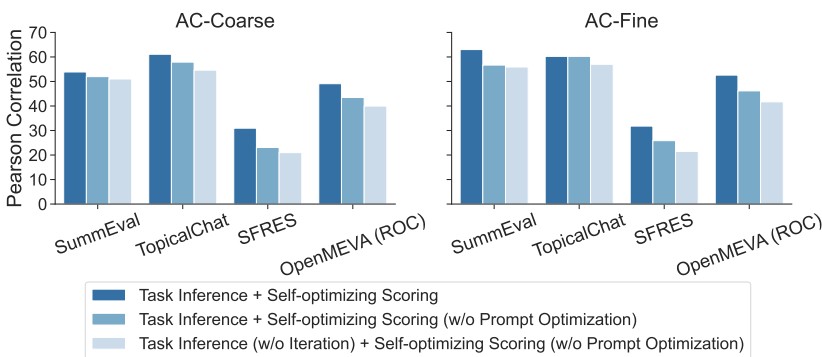

Figure 6: Effectiveness of Optimization. We report the Pearson ($\gamma$) correlation coefficient for our two optimal experimental variants: AC-Coarse and AC-Fine.

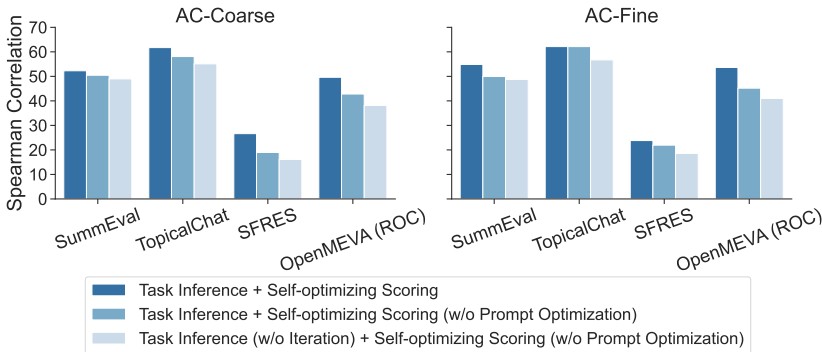

Figure 7: Effectiveness of Optimization. We report the Spearman ($\rho$) correlation coefficient for our two optimal experimental variants: AC-Coarse and AC-Fine.

## F   DETAILS OF HUMAN EVALUATION

# Human Eval for Explainations

I will provide you with instances from the SummEval dataset, each randomly selected and categorized into three score ranges: 0-50, 51-80, and 81-100, with 10 instances per category. Each instance includes a detailed evaluation of a summary response to a source text. The evaluation covers several dimensions: coherence, conciseness, coverage, accuracy, fluency, relevance, clarity, and engagement, accompanied by detailed explanations and scores for each. The overall quality is also assessed.

Your task is to **assess the explanations in these instances using the provided criteria below**. Please begin your evaluation now. Keep the document open at all times and consult it as necessary to guide your assessment of the specific evaluation criteria.

**Instance Number**

Copy the instance number, for example, (0-50)_1

_______________________________________

►Please read the explanation for each dimension in '**Explanation**' carefully, and judge whether each explanation is unambiguous and easy to understand.

**Clarity:** Is the explanation unambiguous and easy to understand?

**Yes**: The explanation is concise, clear, and free of confusing terminology or expressions.

**No**: The explanation contains ambiguity or confusing terms that make it hard to understand.

|  | Yes | No |
|---|---|---|
| Coherence | ○ | ○ |
| Conciseness | ○ | ○ |
| Coverage | ○ | ○ |
| Accuracy | ○ | ○ |
| Fluency | ○ | ○ |
| Relevance | ○ | ○ |
| Clarity | ○ | ○ |
| Engagement | ○ | ○ |
| Overall Quality | ○ | ○ |

►Please read the explanation for each dimension in '**Explanation**' carefully, and judge whether each explanation reflects and closely relates to its evaluation dimension.

**Relevance**: Does the explanation accurately reflect and closely relate to its evaluation dimension?

**Yes:** The explanation accurately reflects and closely relates to the evaluation dimension.

**No:** The explanation does not accurately reflect or closely relate to the evaluation dimension.

|  | Yes | No |
|---|---|---|
| Coherence | ○ | ○ |
| Conciseness | ○ | ○ |
| Coverage | ○ | ○ |
| Accuracy | ○ | ○ |
| Fluency | ○ | ○ |
| Relevance | ○ | ○ |
| Clarity | ○ | ○ |
| Engagement | ○ | ○ |
| Overall Quality | ○ | ○ |

►Please read the explanation and score for each dimension in '**Explanation**' carefully, and judge whether each explanation reflects the assigned score.

**Explanation and Score Alignment**: Does the explanation appropriately reflect the assigned score, and can the user understand the reason for the assigned score through the explanation?

**Yes:** The explanation content clearly reflects the assigned score, and the user can understand the reason for the score.

**No:** The explanation content does not clearly reflect the assigned score, and the user cannot understand the reason for the score.

|  | Yes | No |
|---|---|---|
| Coherence | ○ | ○ |
| Conciseness | ○ | ○ |
| Coverage | ○ | ○ |
| Accuracy | ○ | ○ |

| | | |
|---|---|---|
| Fluency | ○ | ○ |
| Relevance | ○ | ○ |
| Clarity | ○ | ○ |
| Engagement | ○ | ○ |
| Overall Quality | ○ | ○ |

►Please read the '**Source**' and '**Explanation**' carefully, and judge whether each explanation matches the source.

**Accuracy**: Does the explanation match the source?

**Yes:** The explanation matches the source text, accurately reflecting the source data or facts, with no hallucinations.

**No:** The explanation does not match the source, containing inaccuracies or hallucinations.

| | Yes | No |
|---|---|---|
| Coherence | ○ | ○ |
| Conciseness | ○ | ○ |
| Coverage | ○ | ○ |
| Accuracy | ○ | ○ |
| Fluency | ○ | ○ |
| Relevance | ○ | ○ |
| Clarity | ○ | ○ |
| Engagement | ○ | ○ |
| Overall Quality | ○ | ○ |

►Please read the '**Explanation**' carefully and judge from an overall perspective whether the overall explanation aligns with the explanations for each dimension.

**Overall Alignment**: Does the overall explanation align with the explanations for each dimension?

**Yes:** The overall explanation is consistent with each dimension's explanation and avoids any contradictory meanings.

**No:** The overall explanation is inconsistent with the explanations for each dimension and contains contradictory meanings.

|  | Yes | No |
|---|---|---|
| Overall Alignment | ○ | ○ |

►Please read the '**Explanation**' carefully and judge from an overall perspective whether the explanation clearly differentiates the current score segment from others.

**Score Segment Differentiation**: Does the explanation clearly differentiate the current score segment from others?

**Yes:** The explanation shows the unique characteristics of its score segment and distinguishes it from other segments, ensuring clear and transparent scoring.

**No:** The explanation does not clearly show the unique traits of its score segment and fails to distinguish it from other segments, which may cause confusion in scoring.

|  | Yes | No |
|---|---|---|
| Overall Alignment | ○ | ○ |

**Overall**: Review all your previous evaluations and give an overall score for the explanation text in the current instance.

○**1:** Very poor quality, most aspects need significant improvement.

○**2:** Poor quality, several key aspects need improvement.

○**3:** Average quality, some aspects are good, but others need improvement.

○**4:** Good quality, most aspects meet standards with minor improvements needed.

○**5:** Excellent quality, all aspects are outstanding and consistent.

