# OpenReview forum: "Large Language Models Are Active Critics in NLG Evaluation"
_ICLR.cc/2025/Conference — Submitted to ICLR 2025_

### Official Review · Reviewer_LNFu · 2024-10-20

**Soundness:** 3
**Presentation:** 3
**Contribution:** 2
**Rating:** 6
**Confidence:** 4

**Summary:**

This paper studies employing LLMs for NLG evaluation. Existing NLG evaluation approaches are shaped by the developers’ understanding and expectations of the evaluation process as “passive critics”. This paper proposes active critics that automatically infer what and how to evaluate from a few labeled data. This framework works as a two-stage pipeline that (1) infers the target NLG task and identify relevant evaluation criteria from the data, and (2) optimize prompts to make its scoring better align with human judgments.

**Strengths:**

1. This paper studies an important research question of reference-free NLG evaluation.

2. The active evaluator proposed in this work exhibits better generalization compared to existing passive NLG evaluators.

**Weaknesses:**

1. Dependence on human annotations. During the first stage of *task inference*, this module is asked to prompt the LLM to formulate an accurate task description by reviewing examples in the training set and identify key information that characterizes the target NLG task. As illustrated in the left side of Figure 1, a human score should be provided for each example. I am curious why we need to go through a complicated process to get the task description when there are already human annotations. We can evaluate the model to align directly with the human annotations. Can you clarify the specific advantages of your approach compared to directly using human annotations?

2. How many training examples should be used for *task inference*? From Section 4.1, it seems like all examples in the training set are split into mini-batches and feed them into LLMs for *task inference*. If only a small amount of training data is needed, then this is acceptable when adapting to a new task. However, if a lot of data, or even the whole dataset, is needed to get the task description, the demand for manual annotation is huge, which is unacceptable.

3. The definition of *comprehensiveness*. In line 206, to enhance efficiency, LLMs are instructed to decide whether to stop early based on the comprehensiveness of the generated task description and criteria set after processing each mini-batch. But how to decide the comprehensiveness is not explained. Can you provide a clear definition or metric for how comprehensiveness is measured and how the early stopping decision is made.

4. A large number of evaluator calls for a single evaluation example. Iterative task inference estimation based on comprehensiveness, prompt optimization for scoring, criterion scoring with explanation for each of the criteria, and an overall scoring to summarize all these criteria. The cost of calling a large number of LLM evaluator is also high, which should also be reported. Can you provide the average number of LLM calls per evaluation and compare these to baseline methods. If there are potential optimizations to reduce the number of LLM calls?

**Questions:**

See above.

---

> ### Author Response · Authors · 2024-11-24
> **Response to Reviewer LNFu**
>
> **Q1: Dependence on human annotations.**
>
> Thank you for raising this insightful concern about our approach to deriving task descriptions in the context of NLG evaluation. The existing paradigm primarily depends on human experts to craft task descriptions and evaluation criteria as the foundation for prompt design.
>
> However， LLMs are highly sensitive to the nuances of task descriptions, and even minor variations in wording can significantly influence the model's outputs. Human annotators, while invaluable, may not consistently capture the subtlety required in task descriptions across varied and intricate datasets. This inconsistency can be problematic, especially given that human-defined descriptions often require extensive trial and error to refine, making the process time-consuming and less scalable.
>
> By leveraging LLMs to infer task descriptions from a limited set of human-scored examples, our method significantly reduces human labor. The LLM iteratively refines and adjusts these descriptions based on outcomes and data, which minimizes the need for broad human intervention. This not only enhances consistency and precision in task definitions but also ensures that the model can adapt quickly to different data characteristics.
>
> Moreover, in practical applications where datasets may involve interlinked tasks, manually generating precise task descriptions becomes impractical. Machines, however, can manage this complexity more efficiently, producing optimized descriptions that cater to specific tasks while maintaining a holistic understanding of the dataset.
>
> In summary, while direct human annotation remains crucial, integrating model-generated task descriptions enables a scalable, efficient, and precise approach. This strategy conserves resources and improves the model's ability to perform accurately and consistently across various tasks, adapting dynamically to the evolving needs of real-world applications.
>
> **Q2: Training examples for task inference.**
>
> Thank you for your concern regarding the training data size for task inference. We keep the annotation requirements manageable and cost-effective. We would like to emphasize that our reliance on human-rated data is minimal, requiring only 25% of the available dataset for LLM adaptation. This is significantly less demanding than existing methods involving fine-tuning an LLM for evaluation, which require order-of-magnitude more training data.
>
> Furthermore, we have set up an early stopping mechanism during the task inference stage. This mechanism allows the LLM to stop reviewing all available data once the LLM is able to reliably infer the task description. As a result, we observe that the model often requires much fewer data points than 25% of the dataset. Specifically, the model uses 60 examples for SummEval, 60 for TopicalChat, 100 for SFRES, and 125 for OpenMEVA, ensuring the process remains efficient and within practical limits.
>
> **Q3: Definition of comprehensiveness and early stopping decision.**
>
> Definition of comprehensiveness: As detailed in Table 9 (Appendix A.5), we define comprehensiveness based on the convergence of output across iterations within the task description's prompt template. The LLM determines comprehensiveness by identifying if the current output converges with the previous iteration's output, indicated by the output keyword ‘COMPLETE’.
>
> Early Stopping Decision: Early stopping is triggered when the LLM outputs "COMPLETE" five times consecutively, or when it reaches the maximum limit of 25 iterations, whichever comes first.
>
> **Q4: Number of LLM call for a single evaluation example.**
>
> Thanks for your insightful comment. In the SummEval dataset, our method requires an average of 22 LLM calls per example during the training phase, with associated costs of about $0.075 per example on GPT-3.5.
>
> During the testing phase, each example requires just 2 LLM calls, costing $0.0053 on GPT-3.5. While our method increases consumption during the training phase, these costs are acceptable given its efficiency in the testing phase.
>
> Comparatively, the G-eval method involves 4 calls per example in inference, whereas our method shows superior performance (\~0.5) with fewer calls in the testing phase.
>
> To further reduce LLM calls, we are exploring enhancements in prompt generation, intelligent data sampling, and caching mechanisms. Thank you again for your valuable suggestions.

---

> > ### Comment · Reviewer_LNFu · 2024-11-25
> >
> > Thanks for the response.
> >
> > The response regarding Q2, Q3, and Q4 has addressed my concerns. I think these details are important which should be elaborated in main text for easy understanding.
> >
> > For Q1, I can get a little bit why the proposed method is better than directly aligning to human annotations. But this still cannot convince me, which I think should be explained more to help readers to understand the contributions of this work.
> >
> > Based on the above, I will increase my score.

---

> ### Author Response · Authors · 2024-11-26
> **Reply to Reviewer LNFu for the follow-up discussion of Q1**
>
> We are glad that our response has addressed some of your concerns, and we are very grateful that you have updated your review and the overall assessment score. Regarding Q1, especially for directly using human ratings for evaluation, we would like to provide more detailed explanations below.
>
> Following prior work [1], a major concern with guiding the machine to align with human ratings directly is the potential lack of interpretability in the evaluation process. Specifically, it can be challenging to understand the model's reasoning and assess its reliability. A natural question arises: “To what extent can different LLMs be trusted for reliable evaluation?” [1]. Even worse, the model's final assessment may align with human preferences by chance, while its underlying reasoning process could be incorrect.
>
> To address this concern, we designed Active-Critic to actively involve the LLM in the reasoning process when generating scores. This approach provides more transparency into the evaluation dimensions and specific aspects the model considers, enhancing the interpretability of its evaluation capabilities. By revealing the rationale and key factors behind the grading, Active-Critic helps uncover the model's evaluative reasoning.
>
> Our experimental results, shown in Table 1, further confirm the effectiveness of our approach. When relying solely on human rating alignment (referred to as AC-vanilla), the empirical performance is lower than our AC-Fine, which guides the LLM in reasoning through the evaluation rationales. The intuition behind our method is that LLMs excel in understanding text, and by leveraging their inductive biases and language comprehension, we can create reasoning processes in natural language that more effectively mirror human reasoning. This not only improves evaluation quality and reliability but also brings the evaluation process closer to the nuanced, interpretive nature of human judgments.
>
> Hopefully, our above explanations could address your concerns. If you have any additional questions or need further clarification, we are more than happy to engage in additional discussions.
>
> [1] Li, M., Liu, Z., Deng, S., Joty, S., Chen, N. F., & Kan, M. Y. (2024). Decompose and Aggregate: A Step-by-Step Interpretable Evaluation Framework. arXiv preprint arXiv:2405.15329.

---

### Official Review · Reviewer_WUeR · 2024-10-30

**Soundness:** 3
**Presentation:** 4
**Contribution:** 2
**Rating:** 5
**Confidence:** 4

**Summary:**

This paper introduces ACTIVE-CRITIC, a protocol that enhances the role of large language models as evaluators in natural language generation tasks. Unlike traditional "passive" evaluation systems that rely on predefined criteria, ACTIVE-CRITIC allows LLMs to actively infer task requirements and optimize prompts for human-aligned scoring. Experiments across multiple NLG tasks demonstrate its effectiveness, offering more adaptable and human-like evaluations through a two-stage, criteria-based scoring process.

**Strengths:**

1. The paper presents a straightforward approach that is easy to understand and follow.
2. Addressing NLG evaluation is essential, especially as LLMs continue to evolve, making evaluation standards increasingly important.
3. The two-stage design in ACTIVE-CRITIC enhances explainability and flexibility, allowing for better alignment with human judgments without extensive manual prompt engineering.

**Weaknesses:**

1. The proposed approach is largely a pipeline with prompt engineering. While it enables LLMs to determine evaluation aspects independently, this method appears as a direct and relatively simple extension of prompt engineering, rather than a novel framework.
2. NLG evaluation aims to approximate human judgment, with scoring criteria derived from human-annotated datasets. The paper’s method, which uses LLMs to learn and generate these criteria, may seem contradictory, as it implies that while human scores are considered "gold," human-derived standards are not.
3. For new tasks, collecting human-annotated scores may be more challenging than developing scoring criteria. This raises concerns about the motivation behind prioritizing criterion generation over human annotation.
4. Experiments and comparisons are limited to a few tasks and baselines.

**Questions:**

Could you provide a demonstration of criterion generation on a novel task and the outcomes it yields?

---

> ### Author Response · Authors · 2024-11-24
> **Response to Reviewer WUeR (Part 1 of 2)**
>
> **Q1: The proposed approach novelty.**
>
> Thanks for sharing your thoughtful concerns about the novelty of our proposed approach. We would like to clarify that the primary focus of our study is to introduce a novel LLM-based evaluation framework for the flexible assessment of various NLG tasks. Our approach presents a fundamental shift of the LLM evaluator from a passive critic to an active critic in the evaluation process. While our method involves designing a prompting-based pipeline to enable active evaluations, it fundamentally differs from traditional prompt engineering methods in NLG evaluation. Unlike conventional prompt engineering, where humans manually craft static natural language prompts to guide machines in performing specific tasks, our approach enables LLMs to dynamically generate prompts that adapt flexibly to specific evaluation scenarios. These scenarios involve LLM-inferred descriptions of the target evaluation tasks and the preferred aspects of end users  on the data quality. Additionally, as mentioned by Reviewer DscM’s, one of our work’s strengths is that "automatically designing the NLG evaluation prompts seems to be quite novel and clearly a highly practical and useful application." Our adaptive mechanism is distinct in its ability to go beyond static prompt engineering, offering a more interactive and context-aware evaluation framework.
>
> **Q2: Contradiction in using machine-derived criteria.**
>
> Thank you for raising this insightful concern about the motivation behind our study. We fully agree that the goal of NLG evaluation is to approximate human judgments, and eliciting the rationale behind these judgments is crucial for generating meaningful evaluation criteria that machines can leverage.
>
> The existing paradigm typically relies on human experts to define evaluation criteria—key aspects that evaluators focus on to assess the quality of system outputs—and asks human annotators to either adhere to these predefined criteria or provide holistic assessments of the generated text. However, prior work has shown that annotators often apply nuanced, implicit evaluation criteria that extend beyond the pre-defined ones provided by system developers, and some of these criteria may even conflict with the developers' intended guidelines [1,2,3]. With this concern in mind, we propose to guide an LLM to review human-annotated data, focusing on overall quality, to uncover more nuanced criteria that are critical for a comprehensive understanding of the text quality of individual instances.
>
> To further examine the quality of machine-generated evaluation criteria, we conducted a qualitative analysis of the evaluation criteria generated by Active-Critic, comparing them with the expert-defined criteria from the SummEval dataset (provided in Appendix C). We observed that our approach produced more nuanced, task-specific evaluation criteria, complementing those outlined by NLG system developers. Additionally, each criterion is clearly articulated in text, highlighting its unique characteristics.
>
> [1] Liu Y, Yang T, Huang S, et al. HD-Eval: Aligning Large Language Model Evaluators Through Hierarchical Criteria Decomposition[J]. arXiv preprint arXiv:2402.15754, 2024.
>
> [2] Clark E, August T, Serrano S, et al. All That’s ‘Human’Is Not Gold: Evaluating Human Evaluation of Generated Text[C]//Proceedings of the 59th Annual Meeting of the Association for Computational Linguistics and the 11th International Joint Conference on Natural Language Processing (Volume 1: Long Papers). 2021: 7282-7296.
>
> [3] Celikyilmaz A, Clark E, Gao J. Evaluation of text generation: A survey[J]. arXiv preprint arXiv:2006.14799, 2020.
>
>
> **Q3: Difficulty of collecting human judgements on the data of new tasks.**
>
> Thank you for raising this important point. We share your concern regarding the high cost of human annotations, particularly for new tasks. As noted in the response above, our approach focuses on instructing the LLM to infer evaluation criteria based on the human ratings of overall quality scores, where "the exact evaluation criteria are left to the discretion of the evaluator" [1]. Compared to criterion-specific ratings, we believe this type of annotation can significantly reduce the effort required from annotators. Furthermore, our approach requires only a minimal amount of human-rated data (at most 400 instances across four NLG tasks in our experiments). The primary goal of our method design is to make the collection of limited human annotations more affordable, enabling the LLM to take the lead in the evaluation process.
>
> [1] Clark E, August T, Serrano S, et al. All that's' human'is not gold: Evaluating human evaluation of generated text[J]. arXiv preprint arXiv:2107.00061, 2021.

---

> ### Author Response · Authors · 2024-11-24
> **Response to Reviewer WUeR (Part 2 of 2)**
>
> **Q4: Experiments and comparisons are limited to a few tasks and baselines.**
>
> Thank you for your comment. Following the experimental design of G-Eval, UniEval, GPTScore and TigerScore, we selected four widely recognized datasets: SummEval, TopicalChat, SFRES, and OpenMEVA. These datasets are commonly used in the evaluation of LLM-based systems and span a diverse range of NLG evaluation scenarios, which allow us to robustly test the applicability of our method across different contexts.
>
> Our choice of these specific tasks is guided by two main principles: **(1) Comparability:** they are commonly utilized in prior research, ensuring comparability of our results with existing studies, and **(2) Diversity:** they represent a diverse span of NLG tasks, which helps in demonstrating the versatility of our approach.
>
> Following your helpful suggestions, we compare our approach with two additional latest baselines, i.e., TIGERScore and InstructScore, both of which are finetune-based evaluators. Besides, we additionally test Active-Critic on LLaMA3-8B and GPT-4 and included GPT-4-based approaches for comparison (as shown bellow), which will be incorporated into Table 2 in the updated manuscript.
>
> | Dataset                 | Summ    | Eval    |    | Topical | Chat   |   | SFRES  |  |  | Open   | MEVA   |  |    |
> |-------------------------|---------|---------|----|---------|--------|---|--------|---|---|--------|--------|--------|-------|
> | Metric               | $\gamma$| $\rho$  | $\tau$ | $\gamma$ | $\rho$  | $\tau$ | $\gamma$ | $\rho$ | $\tau$ | $\gamma$ | $\rho$  | $\tau$ | AVE |
> | TIGERScore                | 0.458   | 0.3694  | 0.2937 | 0.3785  | 0.4401 | 0.3458| 0.1898  | 0.1246 | 0.1075| 0.451  | 0.4413  | 0.3356 |  0.3279 |
> | InstructScore                | 0.3496  | 0.2703  | 0.203  | 0.2691  | 0.2774 | 0.2423| 0.2039  | 0.1502 | 0.133 | 0.2234 | 0.1533  | 0.1295 | 0.2171 |
> | AC-Fine (LLaMA3-8B-Based)   | 0.4907 | 0.4772  | 0.3558 | 0.5351  | 0.532  | 0.3972| 0.2265  | 0.2245 | 0.169 | 0.4229 | 0.4343  | 0.3168 | 0.3818 |
> | AC-Fine (Orca2-13B-Based)     | 0.6301 | 0.5486  | 0.4299 | 0.6023  | 0.6214 | 0.4713| 0.2915  | 0.2501 | 0.1906| 0.5259 | 0.5363  | 0.4109 | 0.4591 |
>
> | Dataset                 | Summ    | Eval    |    | Topical | Chat   |   |   |
> |-------------------------|---------|---------|----|---------|--------|---|--------|
> | Metric               | $\gamma$| $\rho$  | $\tau$ | $\gamma$ | $\rho$  | $\tau$ | AVE |
> | GPT3.5-based	             | 0.453  | 0.385   | 0.292	| 0.5503	| 0.5436 | 0.4231| 0.4412 |
> | GPT4-based	             | 0.5943 |0.5038 | 0.4055 | 0.6659  | 0.656  | 	0.4937 | 0.5532 |
> | AC-Fine (GPT3.5-Based)	   | 0.653  | 0.6016  | 0.4745 | 0.6718  | 0.6703 | 0.5156| 0.5978 |
> | AC-Fine (GPT4-Based)     | 0.6779 | 0.5787  | 0.4565 | 0.7224  | 0.7868 | 0.6157| 0.6397 |
>
> **Q5: Demonstration of criteria generation on a new task.**
>
> Thank you for your insightful comments. To assess a new task, we would need a small dataset that include both source text and response text, each paired with human-assigned quality scores. In our experiments, we generally use between 5% and 25% of the test cases, amounting to roughly 80 to 400 data points. The exact number depends on the complexity of the task.
>
> As detailed in Section 4.1 of the paper, we discuss the task inference stage of the ACTIVE-CRITIC protocol. This crucial stage empowers the LLM to autonomously analyze the input dataset, identify the characteristics of the NLG task, and establish accurate evaluation criteria. This process involves two essential components:
>
> (1)**Task Description Generation:** The LLM automatically examines testing cases organized into mini-batches. It combines these observations to create an accurate and comprehensive task description.
>
> (2)**Criteria Definition:** Next, the LLM thoughtfully defines specific evaluation criteria tailored to the target NLG task. This step is conducted without pre-defined standards, focusing instead on identifying the most relevant evaluation aspects.
>
> To ensure efficiency, an early stopping mechanism is applied when the LLM reviews the available data. Therefore, the model may require fewer data points to complete the data analysis once the task description and criteria are sufficiently clear to the model.
>
> This approach highlights the LLM’s remarkable ability to rapidly adapt to new tasks and generate precise task definitions and evaluation criteria with minimal human intervention.

---

> > ### Comment · Reviewer_WUeR · 2024-11-27
> >
> > Thank you for your detailed response and the additional experiments, which are helpful. I appreciate the effort you have put into clarifying your work and strengthening the empirical evidence.
> >
> > However, I still find the proposed method lacks novelty. It has been explored extensively in prior work that using llm to generate plans or instructions for subsequent stages. Without a clear differentiation or significant innovation, the contribution of this work remains limited in this context.
> > For these reasons, I maintain my original score.

---

> > > ### Author Response · Authors · 2024-11-29
> > >
> > > Thank you for taking the time to read our response and sharing follow-up feedback. We would like to summarize our innovation in three key points:
> > >
> > > (1) **Active vs. Passive critic.** Our approach presents a fundamental shift of the LLM evaluator from a passive critic to an active critic in the evaluation process. Our Active-Critic method our work introduces a fully automatic evaluation protocol to infer what and how to evaluate from minimal human-rated data. In contrast, prior approach can be viewed as a passive critic, where the evaluation process is semi-automatic and depends on expert-defined task contexts or evaluation criteria to guide the machine's assessment. Our method enables LLMs to dynamically generate evaluation prompts that adapt flexibly to specific evaluation scenarios. This shift allows the model to be more flexible and adaptive, particularly when evaluating new or diverse NLG tasks.
> > >
> > > (2)**Data-driven vs. Task-driven.** Our approach guides an LLM to review human-annotated data to uncover more nuanced criteria that are critical for a comprehensive understanding of the text quality of individual instances. In contrast, prior methods primarily focus on generating a single set of predefined evaluation criteria per NLG task. This approach may limit the ability to provide personalized evaluations that adapt to specific user needs or different data characteristics.
> > >
> > > (3)**Prompt Optimization vs. Human-defined/Random Sample.** We have designed an automatic prompt optimization strategy that iteratively selects the most relevant context to refine the prompts. Previous methods often overlook the importance of context in the selection of in-context examples. They either rely on human-defined examples or use random sampling to generate evaluation prompts. Our approach, on the other hand, dynamically selects the optimal prompt for the current task, improving both the quality and relevance of the evaluation process.
> > >
> > > We hope these further explanations address your concerns. If you have any additional questions or need further clarification, we are more than happy to engage in additional discussions.

---

### Official Review · Reviewer_SBMj · 2024-11-01

**Soundness:** 2
**Presentation:** 3
**Contribution:** 2
**Rating:** 3
**Confidence:** 4

**Summary:**

This work proposes an active NLG evaluation method based on LLMs. It requires the model to generate task descriptions and evaluation criteria for different cases independently and uses data with human labels and prompt optimization algorithms to improve demonstration selection in few shots. Additionally, the model provides evaluations across different dimensions and offers explanations for its scores, which are then aggregated into overall evaluation results. The experiments cover common NLG tasks such as summarization, dialogue, data-to-text, and story generation.

**Strengths:**

This work highlights a limitation of current LLM-based NLG evaluation methods: they often rely on specific evaluation instructions to perform passive evaluations. While human-crafted evaluation criteria may intuitively enhance the controllability, they lead to high costs. To address this, this work proposes a method for enabling models to conduct active evaluations and conducts experiments on multiple NLG evaluation tasks.

**Weaknesses:**

Although the concept of active evaluation is meaningful, approaches with similar motivation have been explored in previous works (Liu et al., 2024; Li et al., 2024; Liu et al., 2024). Furthermore, prior research has pointed out that generating explanations along with scores can enhance evaluation performance (Chiang et al., 2023; Chiang et al., 2023).

The proposed method requires some training data to fine-tune prompts, which significantly reduces its generalizability (how would it handle evaluation samples in new tasks or domains?). Additionally, in the experiments, they sample 25% of each dataset as the training set, leaving the rest as the test set. This is unfair compared to other baselines (e.g., Auto-J), which had not seen the corresponding dataset and been tested on all data.

The LLMs used in this paper are somewhat outdated (Orca2 instead of Llama3, GPT-3.5 instead of GPT-4). Moreover, the latest evaluation methods are not included for comparison (Kim et al., 2023; Kim et al., 2024; Hu et al., 2024). And the experimental results only present the overall score performance, whereas prior related work has focused more on performance across individual dimensions, which is also necessary to be experimented.

**References:**

Liu, Yuxuan, et al. "Calibrating LLM-Based Evaluator." Proceedings of the 2024 Joint International Conference on Computational Linguistics, Language Resources and Evaluation (LREC-COLING 2024). 2024.

Li, Minzhi, et al. "Decompose and Aggregate: A Step-by-Step Interpretable Evaluation Framework." arXiv preprint arXiv:2405.15329 (2024).

Liu, Yuxuan, et al. "HD-Eval: Aligning Large Language Model Evaluators Through Hierarchical Criteria Decomposition." arXiv preprint arXiv:2402.15754 (2024).

Chiang, Cheng-Han, and Hung-Yi Lee. "Can Large Language Models Be an Alternative to Human Evaluations?." Proceedings of the 61st Annual Meeting of the Association for Computational Linguistics (Volume 1: Long Papers). 2023.

Chiang, Cheng-Han, and Hung-yi Lee. "A closer look into automatic evaluation using large language models." arXiv preprint arXiv:2310.05657 (2023).

Kim, Seungone, et al. "Prometheus: Inducing fine-grained evaluation capability in language models." The Twelfth International Conference on Learning Representations. 2023.

Kim, Seungone, et al. "Prometheus 2: An open source language model specialized in evaluating other language models." arXiv preprint arXiv:2405.01535 (2024).

Hu, Xinyu, et al. "Themis: Towards flexible and interpretable nlg evaluation." arXiv preprint arXiv:2406.18365 (2024).

**Questions:**

Please refer to Weaknesses.

---

> ### Author Response · Authors · 2024-11-24
> **Response to Reviewer SBMj (Part 1 of 3)**
>
> **Q1: Comparison with Previous Works (Liu et al., 2024; Li et al., 2024; Liu et al., 2024) and Our Main Contribution.**
>
> Thank you for highlighting these relevant works. We agree that the listed latest works (Liu et al., 2024a; Li et al., 2024; Liu et al., 2024b), similar to ours, focus on guiding the LLM to automatically evaluate NLG systems. Despite that, we still find several fundamental differences in motivation and/or methodology design between these studies and ours:
>
> ● **Manually Predefined Criteria vs Automatic Criteria Generation**: Our work introduces an approach that instructs an LLM to automatically generate evaluation criteria by analyzing a small dataset with early stopping (Section 4.1), significantly reducing the intensive human effort required to manually define task-specific evaluation criteria based on random in-context examples, as done in Liu et al. (2024a).
>
> ● **Manually Predefined Task Information v.s. Automatic Task Understanding**: Our work proposes to instruct an LLM to automatically analyze the NLG task by reviewing valuable data contributed by end users of NLG systems. These users are uniquely positioned to identify nuanced errors in system outputs during deployment. In contrast, prior studies, such as Li et al. (2024), rely on the NLG system developers to manually predefine the task context within the prompt, which may potentially overlook the realistic demands and expectations of end-users. Moreover, as noted in the appendix of Li et al. (2024), their method adopts the same prompting templates from the original experiment of each benchmark for the direct scoring method as they are carefully designed for the specific requirement of each task. Our framework uses a generic prompt, removing such a requirement.
>
> ● **Rigid Task-specific Criteria vs. Adaptive Data-driven Criteria**: Prior studies, such as Lie et al. (2024b), typically use a standard set of evaluation criteria for each NLG task. However, we argue that even within the same NLG task, such as text summarization, the evaluation criteria should vary depending on the text domains (e.g., news articles v.s. Scientific papers). To overcome these limitations, we propose a novel data-driven approach to adaptively identify the nuanced criteria that truly matter to end users.
>
> We appreciate your insights and pointers to prior works highlighting the benefits of explanations. However, we would like to clarify that our primary focus is the proposal of a new LLM-based NLG evaluation paradigm centered on active critics. Our goal is to develop an adaptable NLG evaluation protocol that can easily apply to various NLG tasks, offering more flexible, personalized, and interpretable evaluation in support of the diverse needs of end users. Our proposed active critic method uses a backbone LLM to actively search for the most relevant evaluation criteria from data, automatically optimize in-context examples for evaluation, and generate text explanations of the evaluation process, harnessing the LLM’s impressive text generation capability.
>
> **Q2: The process of generalizing to a new task.**
>
> Thank you for highlighting this important concern. Our approach uniquely enables the automatic generation of evaluation task definitions and criteria, distinguishing itself from traditional methods that require human-crafted definitions for each specific task. This innovative approach significantly reduces the manual effort required to design detailed task definitions, allowing the model to adapt more flexibly to new domains or tasks.
>
> Regarding the use of labeled data, we have strategically employed a lightweight data-driven approach. For instance, in our experiments, we use only 5% to 25% of the dataset as the training set, which contains approximately 80 to 400 data points. Depending on the complexity of the NLG task, these data points can be labeled within a reasonable timeframe.  This subset has been found to be sufficient for fine-tuning the prompts while still maintaining the method’s robust generalization capabilities.
>
> When addressing a new task, such as assessing cultural awareness in storytelling, our method efficiently utilizes a small set of stories coupled with a general human evaluation of cultural awareness in each story. This streamlined process eliminates the need for detailed initial data, allowing the language model to autonomously generate precise task definitions and crucial aspects related to cultural awareness for subsequent evaluation. We believe this demonstrates the method’s practical applicability and adaptability to diverse evaluation contexts.

---

> ### Author Response · Authors · 2024-11-24
> **Response to Reviewer SBMj (Part 2 of 3)**
>
> **Q3: Concerns About Fairness in Comparisons with Other Baselines.**
>
> Thank you for your concerns regarding the fairness of our comparisons with other baselines. We would like to clarify that our experimental setup, as detailed in our paper, ensured that both our method and the baselines were evaluated under identical conditions. Specifically, all methods were tested on the same dataset, which constitutes 75% of the total data.
>
> Furthermore, we would like to highlight that the baselines we compared against, such as Auto-J, InstructScore, and TIGERScore, are trained on diverse corpora designed for various tasks, including text summarization. These baselines adhere to clearly defined and established criteria.
>
> With these points in mind, we firmly believe that our comparative analysis is conducted with fairness and rigor.
>
> **Q4: Concerns over outdated LLMs (Orca2-13B and GPT3.5) and lack of lasted evaluation methods in Active-Critic assessment.**
>
> We appreciate your insightful comment on the robustness of our Active-Critic with different LLM backbones. We do understand this conscern and hence we have employed two base LLMs, an open-source model (ORCA-13B, released on November 23, 2023) and a stronger closed-source one (GPT3.5, released on November 6, 2023), for our evaluation protocal assessment. Following your helpful suggestions, we additionally test Active-Critic on LLaMA3.2-8B and GPT-4. Besides, we compare our approach with two additional latest baselines, i.e., TIGERScore (Jiang et al., 2023) and InstructScore (Xu et al., 2023), both of which are finetune-based evaluators. The following results show that our Active-Critic approach outperforms both TIGERScore and InstructScore across four distinct NLG tasks, achieving further improvements when utilizing stronger backbone LLMs like GPT-3.5 and GPT-4.
>
> | Dataset                 | Summ    | Eval    |    | Topical | Chat   |   | SFRES  |  |  | Open   | MEVA   |  |    |
> |-------------------------|---------|---------|----|---------|--------|---|--------|---|---|--------|--------|--------|-------|
> | Metric               | $\gamma$| $\rho$  | $\tau$ | $\gamma$ | $\rho$  | $\tau$ | $\gamma$ | $\rho$ | $\tau$ | $\gamma$ | $\rho$  | $\tau$ | AVE |
> | TIGERScore                | 0.458   | 0.3694  | 0.2937 | 0.3785  | 0.4401 | 0.3458| 0.1898  | 0.1246 | 0.1075| 0.451  | 0.4413  | 0.3356 |  0.3279 |
> | InstructScore                | 0.3496  | 0.2703  | 0.203  | 0.2691  | 0.2774 | 0.2423| 0.2039  | 0.1502 | 0.133 | 0.2234 | 0.1533  | 0.1295 | 0.2171 |
> | AC-Fine (LLaMA3-8B-Based)   | 0.4907 | 0.4772  | 0.3558 | 0.5351  | 0.532  | 0.3972| 0.2265  | 0.2245 | 0.169 | 0.4229 | 0.4343  | 0.3168 | 0.3818 |
> | AC-Fine (Orca2-13B-Based)     | 0.6301 | 0.5486  | 0.4299 | 0.6023  | 0.6214 | 0.4713| 0.2915  | 0.2501 | 0.1906| 0.5259 | 0.5363  | 0.4109 | 0.4591 |
>
> | Dataset                 | Summ    | Eval    |    | Topical | Chat   |   |   |
> |-------------------------|---------|---------|----|---------|--------|---|--------|
> | Metric               | $\gamma$| $\rho$  | $\tau$ | $\gamma$ | $\rho$  | $\tau$ | AVE |
> | GPT3.5-based	             | 0.453  | 0.385   | 0.292	| 0.5503	| 0.5436 | 0.4231| 0.4412 |
> | GPT4-based	             | 0.5943 |0.5038 | 0.4055 | 0.6659  | 0.656  | 	0.4937 | 0.5532 |
> | AC-Fine (GPT3.5-Based)	   | 0.653  | 0.6016  | 0.4745 | 0.6718  | 0.6703 | 0.5156| 0.5978 |
> | AC-Fine (GPT4-Based)     | 0.6779 | 0.5787  | 0.4565 | 0.7224  | 0.7868 | 0.6157| 0.6397 |

---

> ### Author Response · Authors · 2024-11-24
> **Response to Reviewer SBMj (Part 3 of 3)**
>
> Furthermore, we have compared our Active-Critic method with the Themis (Hu et al., 2024) you mentioned (results shown below). Our results show that our Active-Critic (GPT-4 backbone) achieves performance comparable to Themis. While Themis has made innovative strides, our approach offers several advantages:
>
> (1)**Minimal Labeled Data Requirement:** Our method requires only 25% of the dataset for effective training, which translates to a maximum of 400 samples in the SummEval dataset. In contrast, Themis requires 0.5 million samples to train their model, resulting in significant time and manpower costs.
>
> (2)**Dynamic and Adaptive Generalization:** Active-Critic dynamically adapts to the latest NLG tasks. While Themis trains on a static dataset and has shown potential to generalize to unseen tasks, its ability to adapt to completely new tasks as the NLG field evolves remains to be proven. Our method, however, can continually improve and adapt to new tasks with minimal human effort.
>
> (3)**Automatic Criteria Inference:** Our approach enables the LLM to actively infer key evaluation criteria, significantly reducing the human effort needed for prompt creation, task analysis, and criteria definition. Themis, on the other hand, relies on human-predefined task definitions or evaluation criteria.
>
> These distinctions highlight our method's capability to not only keep pace with rapid advancements in the field of NLG but also to reduce the operational demands typically associated with static and extensive training data requirements.
>
> We also thank you for sharing the other methods (Kim et al., 2023; Kim et al., 2024). We will supplement the experiments as soon as possible and add them to the revised version.
>
> | Dataset                 | Summ    | Eval    |    | Topical | Chat   |   |   |
> |-------------------------|---------|---------|----|---------|--------|---|--------|
> | Metric               | $\gamma$| $\rho$  | $\tau$ | $\gamma$ | $\rho$  | $\tau$ | AVE|
> | Themis                   | 0.6134  |	0.545 |0.4765 |	0.7452  |0.716    |0.6429 | AVE|
> | AC-Fine (GPT4-Based)     | 0.6779 | 0.5787  | 0.4565 | 0.7224  | 0.7868 | 0.6157| AVE|
>
> Jiang, Dongfu, et al. "Tigerscore: Towards building explainable metric for all text generation tasks." Transactions on Machine Learning Research (2023).
>
> Xu, W., Wang, D., Pan, L., Song, Z., Freitag, M., Wang, W. Y., & Li, L. (2023). INSTRUCTSCORE: Explainable Text Generation Evaluation with Finegrained Feedback. arXiv preprint arXiv:2305.14282.
>
> Hu, Xinyu, et al. "Themis: Towards flexible and interpretable nlg evaluation." arXiv preprint arXiv:2406.18365 (2024).
>
> **Q5: Performance across indicidual dimensions.**
>
> Thanks for your insightful comments. Referring to the existing works [1, 2, 3, 4], machine-based evaluations can generally be categorized as either criteria-specific or providing an overall rating. Our approach falls into the latter category. Additionally, our method introduces nuanced aspects that extend beyond those defined by humans. This presents a challenge in comparing correlations for these aspects, as human ratings are not available. We also wish to highlight that our approach is capable of providing aspect-specific scores. The human evaluations conducted on each aspect affirm that each aspect’s score aligns with human expectations.
>
> [1] Jiang, Dongfu, et al. "Tigerscore: Towards building explainable metric for all text generation tasks." Transactions on Machine Learning Research (2023).
>
> [2] Xu W, Wang D, Pan L, et al. INSTRUCTSCORE: Towards Explainable Text Generation Evaluation with Automatic Feedback[C]//Proceedings of the 2023 Conference on Empirical Methods in Natural Language Processing. 2023: 5967-5994.
>
> [3] Li J, Sun S, Yuan W, et al. Generative Judge for Evaluating Alignment[C]//The Twelfth International Conference on Learning Representations.
>
> [4] Celikyilmaz A, Clark E, Gao J. Evaluation of text generation: A survey[J]. arXiv preprint arXiv:2006.14799, 2020.

---

> ### Comment · Reviewer_SBMj · 2024-11-24
>
> Thank you for your detailed response, but I believe my main concerns have not been addressed:
>
> 1. I understand that your work differs to some extent from other approaches utilizing LLMs for active evaluation, but I think the motivation and methodology are similar. So it is necessary to clearly highlight the innovative contributions of your work and compare the performance of your method to other related methods to demonstrate your practical advantages.
>
> 2. Similar concerns have also been raised by other reviewers, namely that your method requires a certain proportion of in-domain supervised data for training. Firstly, such data also requires human involvement and incurs annotation costs, which somewhat contradicts the intention of reducing human effort in designing evaluation criteria when leveraging LLMs for evaluation. Furthermore, when facing new evaluation tasks, it is typically impossible to obtain immediate supervised data, which directly hinders the applicability of your proposed method. For instance, when evaluating a cover letter, there may not be any existing evaluation dataset or human-annotated data available.
>
> 3. I believe my concerns about fairness were misunderstood. Although methods like Auto-J, InstructScore, and TIGERScore use training data from the same task as the test data, they do not use a subset of the actual test data for training. For example, they might be trained with evaluation data on summarization outside of SummEval but test on the full SummEval dataset. However, your method involves both training and testing on SummEval, which represents a fundamentally different requirement for model generalization. While you report the performance of all methods on the same part of the testset, such comparisons remain unfair.
>
> 4. I appreciate the additional experiments conducted with the latest GPT-4 and Llama-3.2 models and the results presented. However, I am puzzled as to why GPT-4 performs significantly worse than weaker GPT-3.5 on SummEval, which seems counterintuitive.
>
> In summary, I have decided to maintain my original score.

---

> > ### Author Response · Authors · 2024-11-27
> > **Reply to Reviewer SBMj for the follow-up discussion (Part 1 of 2)**
> >
> > Thank you for taking the time to read our response and sharing follow-up considerations. Regarding the key innovation and contribution of our work compared to the listed prior work and empirical comparisons, we would like to share our thoughts below.
> >
> > 1\. **Innovative contributions**: Our work introduces a fully automatic evaluation protocol where the LLM self-infers the target NLG evaluation task, generates evaluation criteria, and creates scoring rubrics (i.e., grading explanations in our case) with minimal human-rated data. Our approach is motivated by two goals: (1) uncovering nuanced criteria beyond the pre-defined ones provided by system developers, as annotators often apply implicit evaluation criteria that extend beyond the pre-defined ones, and some of these criteria may even conflict with the developers' intended guidelines, and (2) enabling flexible, personalized, and interpretable evaluations to address diverse user needs, even for the same task or data. By actively identifying the most relevant evaluation criteria from data, our data-driven strategy empowers the machine to adaptively focus on the criteria that matter most to end users.
> >
> > Comparatively, approaches proposed by Liu et al. (2024a) and Li et al. (2024) can be viewed as a semi-automatic evaluation process, given that they either require expert-defined task contexts or evaluation criteria as input to enable the machine to conduct further calculations. Liu et al. (2024b) primarily focus on generating a single set of evaluation criteria per NLG task, which may limit their approach’s ability to deliver personalized evaluations.
> >
> > **Comparative performance**: We do share your consideration about comprehensive empirical comparisons. Therefore, following your and other reviewers’ helpful suggestions, we provide additional experiments on GPT-4 and LLama-3. Moreover, we compare our methods with additional latest baselines, including TIGERScore, INSTRUCTScore, and Themis. Our results show that our method outperforms existing approaches.
> >
> > We also explored the codebase of Liu et al. (2024a) and Liu et al. (2024b). However, to the best of our knowledge, their codebases are not accessible at this moment (no links are provided in their papers, and couldn’t find the resources via Google Search Engine). Although we could achieve the codebase of Li et al. (2024) on GitHub (https://github.com/WING-NUS), it shows that the code was updated at the beginning of July. Following the policy of ICLR 2025 about concurrent work (https://iclr.cc/Conferences/2025/ReviewerGuide), this work can be viewed as a concurrent one that is not required to compare.
> >
> > 2\. **Human involvement and annotation cost**: We would like to clarify that our primary goal is to reduce, rather than eliminate, human effort in NLG evaluation. Existing methods heavily rely on expert input to define evaluation tasks and criteria. Prompting-based approaches additionally require experts to manually craft task-specific prompts for LLM-based evaluations. Differing from prior approaches, by asking the end users to provide limited holistic ratings of generated texts, we believe our approach could significantly lower the human effort involved. Moreover, we try to minimize the number of human annotations required in our approach—only 25% of the testing data—far less than the extensive data needed for fine-tuning LLMs. This design prioritizes affordability in collecting limited human annotations, enabling the LLM to play a leading role in the evaluation process.
> >
> > **Generation on a new task**: We completely agree that “there may not be any existing evaluation dataset or human-annotated data available” for a new task. However, we assume that when a new task is proposed, the task proposers have curated a dataset for it. We only require the proposers to annotate a minimal number of test cases regarding their overall quality. Specifically, we envision an iterative evaluation paradigm involving humans in the loop. Our method would start with a zero-shot or 5-shot evaluation, and the task developers can gradually correct or confirm the model’s evaluation judgments. Then we can gradually improve the quality and quantity of the available few-shot examples for the next iteration until little is required by human correction. This is an interesting and meaningful idea to explore for LLM evaluation on new NLG tasks, which we believe deserves a dedicated study for this new setting as our future work.

---

> ### Author Response · Authors · 2024-11-27
> **Reply to Reviewer SBMj for the follow-up discussion (Part 2 of 2)**
>
> 3\. **Concerns about fairness**: Following our detailed literature review, we observed that TIGERScore employed full SummEVAL data in their fine-tuning process (as shown in Table 2 in [1]). Auto-J used a relevant dataset, i.e., OpenAI Summary, for training, and INSTRUCTScore prompted GPT4 to synthesize data for fine-tuning. Although Auto-J and INSTRUCTScore did not train on the same SummEval dataset, there may exist data leakage risks where advanced LLMs have access to such data during their pre-training process. In this case, we believe that our comparisons are comparatively fair enough.
>
> [1] Jiang, Dongfu, et al. "Tigerscore: Towards building explainable metric for all text generation tasks." Transactions on Machine Learning Research (2023).
>
> 4\. Thanks for raising this interesting question. To clarify, our experiments primarily focus on comparing the performance of zero-shot GPT 3.5 vs. AC-Fine (GPT 3.5) and zero-shot GPT 4 vs. AC-Fine (GPT 4), to demonstrate the effectiveness of our approach in enhancing advanced closed-source LLMs for NLG evaluations. The results show that our approach noticeably surpasses the zero-shot version of GPT-3.5 and GPT-4, suggesting our strategy of initiating the machine’s active thinking capability is helpful in boosting the base LLMs’ performance in the process of evaluation.
>
> Given the limited information about the pre-training data, evaluation metrics, and optimization processes of these closed-sourced LLMs, we are afraid that a direct comparison based solely on their performance in a single task, e.g., SummEval, may not fully capture the differences in their overall capabilities. We believe this issue is worthy of further exploration in future work.
>
> We hope our explanations address your concerns. If you have any additional questions or need further clarification, we are more than happy to engage in additional discussions.

---

### Official Review · Reviewer_DscM · 2024-11-03

**Soundness:** 3
**Presentation:** 3
**Contribution:** 2
**Rating:** 5
**Confidence:** 3

**Summary:**

When prompting LLMs for NLG evaluation, prompting can be inconvenient due to the required manual intervention to design the prompts, which may also limit the model's ability to leverage other aspects of the text that may be useful for assessment. This paper proposes a method of automatically generating prompts for NLG evaluation. They propose a multi-stage process, where by using batches of test examples with human labels, the LLM first predicts the task of interest, attributes relevant for examples and suitable few-shot examples. This is then used to develop a prompt that can be used for NLG evaluation on new test cases, which to aid assessment, also provides explanations. The work provides ablations for the different components of the framework, demonstrating that for ORCA-13B and ChatGPT-3.5 the approach results in better performance than compared to existing baselines, and that the generated explanations are helpful for evaluation.

**Strengths:**

- As far as I’m aware, I haven’t encountered works that apply prompt optimization for NLG evaluation, and so automatically designing the NLG evaluation prompts seems to be quite novel and clearly a highly practical and useful application. Also their approach decomposes the evaluation into multiple different aspects (e.g. task, criteria, few-shot examples, explainability).

- They provide ablations numbers to make clearer where the performance improvements originate from, showing that all of the task descriptions, assessing individual criteria and having explanations are all helpful. Along with its judgements, the model returns explainable decisions which can be useful for many use-cases.

- The paper is fairly easy to follow, and details of prompts and meta-prompts are provided.

**Weaknesses:**

- The experimental results have not comprehensively demonstrated the advantages of the active-critic approach. The examined models are somewhat limited (ORCA-13B and GPT3.5) and also the baseline compared to appear somewhat limited inconsistent (e.g. G-EVAL not used in Table 1 and Table 2 only compares to G-EVAL). Whether the approach remains equally as effective when using larger more capables, which may possibly be better aligned, is not clear.

-  Is the contribution of the work the simplicity in the approach, i.e. that humans no longer have to design the prompts, or that by using this approach we get better prompts that result in better performance? I’m not sure if any of the baselines directly compare performance to human design prompts, which, as far as I understand, is the main contribution of the paper. This might be what AC-vanilla is, but from what I can tell, AC-vanilla only provides the system with few shot examples.

- The approach does require labelled examples in order to optimize the prompts, which limits some of the advantages of prompting approaches, even when compared to few-shot in-context learning. Especially with the dataset split of 75%, 25%, the approach is now more manually laborious and we require more reliable assessment scores for the assessment task of interest.

**Questions:**

- Why was G-EVAL not compared to in Table 1? The baselines seem a bit inconsistent across models (I understand that for Table 2 the system is resetrected to API calls and may be blackbox, and hence some approaches may be limited)

- When you randomly select 25% of the data for ACTIVE-CRITIC tuning, are these 25% of contexts (i.e. for SummEval do you take 25 of the articles and all 16 of the summaries?) or 25% of all pairs across contexts?

- Out of curiosity, what was the process of developing the meta-prompts provided in Appendix A.5. Was this just manually designed, or was standard prompts used for automatic prompt optimization works, etc?

---

> ### Author Response · Authors · 2024-11-23
> **Response to Reviewer DscM (Part 1 of 2)**
>
> **Q1. Limited backbone models (Orca2-13B and GPT3.5) for Active-Critic assessment.**
>
> We appreciate your insightful comment on the robustness of our Active-Critic with the change of base LLM, especially for the ones with a higher capacity. We do share this concern and hence we employed two base LLMs, an open-source model (Orca2-13B) and a stronger closed-source one (GPT3.5), for our evaluation protocol assessment. As discussed in lines 297-303, we also pre-tested Active-Critic on LLaMA2-13B in the preliminary study, and found that Orca2-13B outperformed LLaMA2-13B. Therefore, we primarily present Orca-13B-based Active-Critic results in the paper. Following your helpful suggestions, we additionally test Active-Critic on LLaMA3-8B and GPT-4. Besides, we compare our approach with two additional latest baselines, i.e., TIGERScore and InstructScore, both of which are finetune-based evaluators. The following results show that Our Active-Critic approach outperforms both TIGERScore and InstructScore across four distinct NLG tasks, achieving further improvements when utilizing stronger backbone LLMs like GPT-3.5 and GPT-4.
>
> | Dataset                 | Summ    | Eval    |    | Topical | Chat   |   | SFRES  |  |  | Open   | MEVA   |  |    |
> |-------------------------|---------|---------|----|---------|--------|---|--------|---|---|--------|--------|--------|-------|
> | Metric               | $\gamma$| $\rho$  | $\tau$ | $\gamma$ | $\rho$  | $\tau$ | $\gamma$ | $\rho$ | $\tau$ | $\gamma$ | $\rho$  | $\tau$ | AVE |
> | InstructScore                | 0.3496  | 0.2703  | 0.203  | 0.2691  | 0.2774 | 0.2423| 0.2039  | 0.1502 | 0.133 | 0.2234 | 0.1533  | 0.1295 | 0.2171 |
> | TIGERScore                | 0.458   | 0.3694  | 0.2937 | 0.3785  | 0.4401 | 0.3458| 0.1898  | 0.1246 | 0.1075| 0.451  | 0.4413  | 0.3356 |  0.3279 |
> | AC-Fine (LLaMA3-8B-Based)   | 0.4907 | 0.4772  | 0.3558 | 0.5351  | 0.532  | 0.3972| 0.2265  | 0.2245 | 0.169 | 0.4229 | 0.4343  | 0.3168 | 0.3818 |
> | AC-Fine (Orca2-13B-Based)     | 0.6301 | 0.5486  | 0.4299 | 0.6023  | 0.6214 | 0.4713| 0.2915  | 0.2501 | 0.1906| 0.5259 | 0.5363  | 0.4109 | 0.4591 |
>
> | Dataset                 | Summ    | Eval    |    | Topical | Chat   |   |   |
> |-------------------------|---------|---------|----|---------|--------|---|--------|
> | Metric               | $\gamma$| $\rho$  | $\tau$ | $\gamma$ | $\rho$  | $\tau$ | AVE |
> | GPT3.5-based	             | 0.453  | 0.385   | 0.292	| 0.5503	| 0.5436 | 0.4231| 0.4412 |
> | GPT4-based	             | 0.5943 |0.5038 | 0.4055 | 0.6659  | 0.656  | 	0.4937 | 0.5532 |
> | AC-Fine (GPT3.5-Based)	   | 0.653  | 0.6016  | 0.4745 | 0.6718  | 0.6703 | 0.5156| 0.5978 |
> | AC-Fine (GPT4-Based)     | 0.6779 | 0.5787  | 0.4565 | 0.7224  | 0.7868 | 0.6157| 0.6397 |
>
> **Q2. Baseline comparison inconsistencies (e.g., G-Eval) when switching from open-source to closed-source backbone LLM.**
>
> Thank you for highlighting this important concern. Our baseline comparisons focus primarily on the type of base LLMs. In Table 1, we compare methods built upon open-source LLMs, while in Table 2, we emphasize comparisons involving closed-source LLM-based methods.  Specifically, Auto-J and UniEval are fine-tuned evaluators, where Auto-J is built upon LLaMA2 and UniEval is built upon T5. ExplainEval is a prompt-based method utilizing Orca2-13B. Following TIGERScore, we built GPTScore-src on FLAN-T5-base. Although the original GPTScore paper included GPT-3.5 as a base LLM, the updates to the GPT-3.5 API make it inapplicable to support GPTScore. As a result, we only compare GPTScore-src in Table 1. Since G-Eval is developed on closed-source LLMs, it is specifically included in Table 2.
>
> While it is possible to adapt G-Eval by using Orca2-13B as the base LLM and compare the modified G-Eval baseline with other methods in Table 1 (results shown below), we are concerned about potential confounding factors that might affect evaluator performance beyond just the change in the base LLM. Additionally, in line with your helpful suggestions in Q1, we have included GPT-4-based approaches for comparison (as shown above), which will be incorporated into Table 2 in the updated manuscript.
> | Dataset                 | Summ    | Eval    |    | Topical | Chat   |   | SFRES  |  |  | Open   | MEVA   |  |    |
> |-------------------------|---------|---------|----|---------|--------|---|--------|---|---|--------|--------|--------|-------|
> | Metric                    | $\gamma$| $\rho$  | $\tau$ | $\gamma$ | $\rho$  | $\tau$ | $\gamma$ | $\rho$ | $\tau$ | $\gamma$ | $\rho$  | $\tau$ | AVE |
> | G-eval (Orca2-13B-Based)      | 0.4549  | 0.4274  | 0.338  | 0.4875  | 0.4512  | 0.3602   | 0.2042  | 0.2066  | 0.146   | 0.4158  | 0.4262 | 0.3007 |0.3516|
> | AC-Fine (Orca2-13B-Based)     | 0.6301 | 0.5486  | 0.4299 | 0.6023  | 0.6214   | 0.4713  | 0.2915  | 0.2501  | 0.1906  | 0.5259  | 0.5363 | 0.4109 |0.4591|

---

> ### Author Response · Authors · 2024-11-23
> **Response to Reviewer DscM (Part 2 of 2)**
>
> **Q3: Approach contribution and baseline comparison regarding the need of human-designed prompts.**
>
> We would like to clarify that our approach eliminates the reliance on manually crafted prompts by enabling the LLM to dynamically generate and adapt prompts based on in-context examples for each target NLG evaluation task. Several of our baselines, including G-Eval, GPTScore-src, and ExplainEval, rely on carefully designed human-created prompts. Our experimental results demonstrate that our approach outperforms these baselines by 0.1404 on average.
>
> While we agree that AC-vanilla also follows a manually crafted prompting strategy, it differs from the aforementioned baselines in that it requires only a few-shot in-context examples, minimizing the human effort needed for prompt design. We include AC-vanilla as a baseline to showcase the intrinsic evaluation capability of the base LLM with minimal manual intervention.
>
> **Q4: Prompt optimization by human-rated data vs. Zero-shot/Few-shot prompting.**
> Thank you for raising this insightful concern. We totally agree with you that existing prompting-based LLM evaluators benefit from minimal supervision on human ratings, enabling them to perform a wide range of evaluation tasks in low-data settings. However, we argue that the applicability of these approaches is contingent on human-defined task specifications and evaluation criteria as input, which require manual analysis by developers or experts. Additionally, the careful design of prompts involves significant human effort and may introduce reliability issues, as LLM performance is highly sensitive to prompt design.
> Our motivation for proposing Active Critic is to address these limitations by enabling the LLM to actively engage with human-rated data, learning from these ratings to infer key evaluation criteria. Unlike existing methods, our approach operates without predefined task definitions or evaluation criteria. This design reduces the need for human effort in prompt creation, task analysis, and criteria definition. However, it does require a small amount of human-rated data as lightweight supervision to help the model infer necessary evaluation materials.
> We would like to emphasize that our reliance on human-rated data is minimal, requiring only 25% of the testing data. This is significantly less demanding than methods involving fine-tuning an LLM for evaluation, which require substantial amounts of training data.
> We also share your concern regarding the reliability of assessments, particularly in relation to data sampling for supervision. In our preliminary analysis, we conducted three rounds of evaluations per task using randomly sampled human-rated data for prompt optimization. As shown in the following example case of TopicalChat, the results demonstrated consistent evaluation outcomes across these rounds (\~0.008 standard deviation on average).
> Building on your consideration, we further examined the impact of labeled data size on the performance of our approach. Specifically, we conducted an analysis on TopicalChat by randomly sampling 5% (18 instances), 15% (54 instances), and 25% (90 instances) of the data for prompt optimization. As shown in the results below, while the average correlation with human ratings improves with larger data sizes for prompt optimization, the differences remain relatively small (\~0.03).
> | Sampling Method (TopicalChat)     | $\gamma$| $\rho$  | $\tau$ | AVE    |
> |-------------------------------------|---------|----------|---------|--------|
> | Randomly Sampling 25% (Round 1) | 0.5849  | 0.5933   | 0.4726  | 0.5503 |
> | Randomly Sampling 25% (Round 2) | 0.6023  | 0.6214   | 0.4713  | 0.565  |
> | Randomly Sampling 25% (Round 3) | 0.6059  | 0.6203   | 0.4746  | 0.5669 |
> | Standard Deviation                 | 0.0092 | 0.013   | 0.0014  | 0.0079 |
>
>
> | Sampling Method (TopicalChat)     | $\gamma$| $\rho$  | $\tau$ | AVE    |
> |-------------------------------------|---------|----------|---------|--------|
> | Randomly Sampling 5%            | 0.531   | 0.5265   | 0.4132  | 0.4902 |
> | Randomly Sampling 15%           | 0.5674  | 0.5836   | 0.4456  | 0.5322 |
> | Randomly Sampling 25%           | 0.6023  | 0.6214   | 0.4713  | 0.565  |
> | Standard Deviation                 | 0.0291 | 0.039   | 0.0238  | 0.0306|
>
> **Q5: How to select training dataset.**
>
> We randomly select 25% of the instance pairs across all contexts. For example, in SummEval, this involves choosing 400 random instances, corresponding to the second method in your description.
>
> **Q6: Development process of meta-prompts in Appendix A.5.**
>
> We followed OpenAI's prompt engineering guidelines (https://platform.openai.com/docs/guides/prompt-engineering) and adapted their well-crafted prompt templates to suit our specific needs. Furthermore, we experimented with various strategies recommended by OpenAI, including role-playing and task decomposition, and identified the most effective approach for our application.

---

### Meta-Review · Area_Chair_QtBo · 2024-12-22

**Metareview:**

After a thorough examination of the paper and the thoughtful feedback provided in the reviewer comments, while I acknowledge this paper addresses an important direction in developing automated NLG evaluation methods with LLMs, there are several significant concerns that need to be addressed. The reviewers consistently point out that the technical novelty is limited, as the approach is largely a straightforward pipeline of prompt engineering techniques, and similar ideas about active evaluation and explanation generation have been explored in previous work. There are also methodological concerns about the experimental evaluation - the comparisons use somewhat outdated models, several important baseline methods are missing, and the results lack comprehensive analysis across individual evaluation dimensions.

**Additional Comments On Reviewer Discussion:**

I noticed there are meaningful dicussions with Reviewer SBMj and the authors and I think I agree with Reviwer SBMj in many aspects.

---

### Decision · Program_Chairs · 2025-01-22

Reject